# PROVABLY LEARNING DIVERSE FEATURES IN MULTI-VIEW DATA WITH MIDPOINT MIXUP

## ABSTRACT

Mixup is a data augmentation technique that relies on training using random convex combinations of data points and their labels. In recent years, Mixup has become a standard primitive used in the training of state-of-the-art image classification models due to its demonstrated benefits over empirical risk minimization with regards to generalization and robustness. In this work, we try to explain some of this success from a feature learning perspective. We focus our attention on classification problems in which each class may have multiple associated features (or *views*) that can be used to predict the class correctly. Our main theoretical results demonstrate that, for a non-trivial class of data distributions with two features per class, training a 2-layer convolutional network using empirical risk minimization can lead to learning only one feature for almost all classes while training with a specific instantiation of Mixup succeeds in learning both features for every class. We also show empirically that these theoretical insights extend to the practical settings of image benchmarks modified to have additional synthetic features.

## 1 INTRODUCTION

Data augmentation techniques have been a mainstay in the training of state-of-the-art models for a wide array of tasks - particularly in the field of computer vision - due to their ability to artificially inflate dataset size and encourage model robustness to various transformations of the data. One such technique that has achieved widespread use is Mixup (Zhang et al., 2018), which constructs new data points as convex combinations of pairs of data points and their labels from the original dataset. Mixup has been shown to empirically improve generalization and robustness when compared to standard training over different model architectures, tasks, and domains (Liang et al., 2018; He et al., 2019; Thulasidasan et al., 2019; Lamb et al., 2019; Arazo et al., 2019; Guo, 2020; Verma et al., 2021b; Wang et al., 2021). It has also found applications to distributed private learning (Huang et al., 2021), learning fair models (Chuang and Mroueh, 2021), semi-supervised learning (Berthelot et al., 2019b; Sohn et al., 2020; Berthelot et al., 2019a), self-supervised (specifically contrastive) learning (Verma et al., 2021a; Lee et al., 2020; Kalantidis et al., 2020), and multi-modal learning (So et al., 2022).

The success of Mixup has instigated several works attempting to theoretically characterize its potential benefits and drawbacks (Guo et al., 2019; Carratino et al., 2020; Zhang et al., 2020; 2021; Chidambaram et al., 2021). These works have focused mainly on analyzing, at a high-level, the beneficial (or detrimental) behaviors encouraged by the Mixup-version of the original empirical loss for a given task.

As such, none of these previous works (to the best of our knowledge) have provided an algorithmic analysis of Mixup training in the context of non-linear models (i.e. neural networks), which is the main use case of Mixup. In this paper, we begin this line of work by theoretically separating the *full training dynamics* of Mixup (with a specific set of hyperparameters) from empirical risk minimization (ERM) for a 2-layer convolutional network (CNN) architecture on a class of data distributions exhibiting a multi-view nature. This multi-view property essentially requires (assuming classification data) that each class in the data is well-correlated with multiple features present in the data.

Our analysis is heavily motivated by the recent work of Allen-Zhu and Li (2021), which showed that this kind of multi-view data can provide a fruitful setting for theoretically understanding the benefits of ensembles and knowledge distillation in the training of deep learning models. We show that Mixup

can, perhaps surprisingly, capture some of the key benefits of ensembles explained by Allen-Zhu and Li (2021) despite only being used to train a single model.

**Main Contributions and Outline.** Our main contributions are three-fold. In Sections 2 and 3, we introduce the main ideas behind Mixup and analyze a simple, linearly separable multi-view data distribution which we use to lay the groundwork for our main results. In analyzing this distribution, we motivate the use of a particular setting of Mixup - which we refer to as Midpoint Mixup - in which training is done on the midpoints of data points and their labels.

Section 4 contains our main results; we prove that, for a highly noisy class of data distributions with two features per class, minimizing the empirical cross-entropy using gradient descent can lead to learning only one of the features in the data while minimizing the Midpoint Mixup cross-entropy succeeds in learning both features. While our theory focuses on the case of two features/views per class to be consistent with Allen-Zhu and Li (2021), our techniques can readily be extended to more general multi-view data distributions.

Lastly, in Section 5, we conduct experiments illustrating that our theoretical insights in Sections 3 and 4 can apply to the training of realistic models on image classification benchmarks. We show for each benchmark that, after modifying the training data to include additional spurious features correlated with the true labels, both Mixup (with standard settings) and Midpoint Mixup outperform ERM on the original test data, with Midpoint Mixup closely approximating the performance of regular Mixup.

**Related Work.** The idea of training on midpoints (or approximate midpoints) is not new; both Guo (2021) and Chidambaram et al. (2021) empirically study settings resembling what we consider in this paper, but they do not develop theory for this kind of training (beyond an information theoretic result in the latter case). As mentioned earlier, there are also several theoretical works analyzing the Mixup formulation and it variants (Carratino et al., 2020; Zhang et al., 2020; 2021; Chidambaram et al., 2021; Park et al., 2022), but none of these works contain optimization results (which are the focus of this work). Additionally, we note that there are many Mixup-like data augmentation techniques and training formulations that are not (immediately) within the scope of the theory developed in this paper. For example, Cut Mix (Yun et al., 2019), Manifold Mixup (Verma et al., 2019), Puzzle Mix (Kim et al., 2020), Co-Mixup (Kim et al., 2021), and Noisy Feature Mixup (Lim et al., 2021) are all such variations.

Our work is also influenced by the existing large body of work theoretically analyzing the benefits of data augmentation (Bishop, 1995; Dao et al., 2019; Wu et al., 2020; Hanin and Sun, 2021; Rajput et al., 2019; Yang et al., 2022; Wang et al., 2022; Chen et al., 2020; Mei et al., 2021). The most relevant such work to ours is the recent work of Shen et al. (2022), which also studies the impact of data augmentation on the learning dynamics of a 2-layer network in a setting motivated by that of Allen-Zhu and Li (2021). However, Midpoint Mixup differs significantly from the data augmentation scheme considered in Shen et al. (2022), and consequently our results and setting are also of a different nature (we stick much more closely to the setting of Allen-Zhu and Li (2021)). As such, our work can be viewed as a parallel thread to that of Shen et al. (2022).

## 2 PRELIMINARIES AND MOTIVATION FOR MIDPOINT MIXUP

We will introduce Mixup in the context of $k$-class classification, although the definitions below easily extend to regression. As a notational convenience, we will use $[k]$ to indicate $\{1, 2, ..., k\}$.

Recall that, given a finite dataset $\mathcal{X} \subset \mathbb{R}^d \times [k]$ with $|X| = N$, we can define the empirical cross-entropy loss $J(g, \mathcal{X})$ of a model $g : \mathbb{R}^d \to \mathbb{R}^k$ as:

$$J(g, \mathcal{X}) = -\frac{1}{N} \sum_{i \in [N]} \log \phi^{y_i}\big(g(x_i)\big) \quad \text{where} \quad \phi^y(g(x)) = \frac{\exp(g^y(x))}{\sum_{s \in [k]} \exp(g^s(x))} \quad (2.1)$$

With $\phi$ being the standard softmax function and the notation $g^y$, $\phi^y$ indicating the $y$-th coordinate functions of $g$ and $\phi$ respectively. Now let us fix a distribution $\mathcal{D}_\lambda$ whose support is contained in $[0, 1]$ and introduce the notation $z_{i,j}(\lambda) = \lambda x_i + (1 - \lambda)x_j$ (using $z_{i,j}$ when $\lambda$ is clear from context) where $(x_i, y_i)$, $(x_j, y_j) \in \mathcal{X}$. Then we may define the Mixup cross-entropy $J_M(g, \mathcal{X}, \mathcal{D}_\lambda)$ as:

$$J_M(g, \mathcal{X}, \mathcal{D}_\lambda) = -\frac{1}{N^2} \sum_{i \in [N]} \sum_{j \in [N]} \mathbb{E}_{\lambda \sim \mathcal{D}_\lambda}\big[\lambda \log \phi^{y_i}(g(z_{i,j})) + (1 - \lambda) \log \phi^{y_j}(g(z_{i,j}))\big] \quad (2.2)$$

We mention a minor differences between Equation 2.2 and the original formulation of Zhang et al. (2018). Zhang et al. (2018) consider the expectation term in Equation 2.2 over $N$ randomly sampled pairs of points from the original dataset $\mathcal{X}$, whereas we explicitly consider mixing all $N^2$ possible pairs of points. This is, however, just to make various parts of our analysis easier to follow - one could also sample $N$ mixed points uniformly, and the analysis would still carry through with an additional high probability qualifier (the important aspect is the *proportions* with which different mixed points show up; i.e. mixing across classes versus mixing within a class).

## 3 MOTIVATING MIDPOINT MIXUP: THE LINEAR REGIME

As can be seen from Equation 2.2, the Mixup cross-entropy $J_M(g, \mathcal{X}, \mathcal{D}_\lambda)$ depends heavily on the choice of mixing distribution $\mathcal{D}_\lambda$. Zhang et al. (2018) took $\mathcal{D}_\lambda$ to be $\mathrm{Beta}(\alpha, \alpha)$ with $\alpha$ being a hyperparameter. In this work, we will specifically be interested in the case of $\alpha \to \infty$, for which the distribution $\mathcal{D}_\lambda$ takes the value $1/2$ with probability 1. We refer to this special case as *Midpoint Mixup*, and note that it can also be viewed as a case of the Pairwise Label Smoothing strategy introduced by Guo (2021). We will write the Midpoint Mixup loss as $J_{MM}(g, \mathcal{X})$ (here $z_{i,j} = (x_i + x_j)/2$ and there is no $\mathcal{D}_\lambda$ dependence as it is deterministic):

$$J_{MM}(g, \mathcal{X}) = -\frac{1}{2N^2} \sum_{i \in [N]} \sum_{j \in [N]} \left( \log \phi^{y_i}(g(z_{i,j})) + \log \phi^{y_j}(g(z_{i,j})) \right) \tag{3.1}$$

We focus on this version of Mixup for a few key reasons. Firstly, we will show that $J_{MM}(g, \mathcal{X})$ exhibits the nice property that its global minimizer corresponds to a model in which all of the features in the data are learned equally (in a sense to be made precise below). We will also show that this is *not* the case for $J_M(g, \mathcal{X}, \mathcal{D}_\lambda)$ when $\mathcal{D}_\lambda$ is *any other* non-trivial distribution. Additionally, from a technical perspective, the Midpoint Mixup loss lends itself to a much cleaner optimization analysis due to the fact that the structure of its gradients is not changing with each optimization iteration (i.e. we do not need to sample new mixing proportions at each step). This allows us to more easily show how the gradient descent dynamics encourage learning all of the features in the data. That being said, we are not trying to claim that Midpoint Mixup is a superior practical alternative to standard Mixup - our goal is simply to show that it better accentuates the theoretical benefits of Mixup, and is empirically comparable to standard Mixup settings. Full proofs for all of the results presented in the next subsection can be found in Section C of the Appendix.

### 3.1 MIDPOINT MIXUP WITH LINEAR MODELS ON LINEARLY SEPARABLE DATA

To make clear what we mean by feature learning, we first turn our attention to the simple setting of learning linear models $g^y(x) = \langle w_y, x \rangle$ (i.e. one weight vector associated per class) on linearly separable data, as this setting will serve as a foundation for our main results. Namely, we consider $k$-class classification with a dataset $\mathcal{X}$ of $N$ labeled data points generated according to the following data distribution (with $N$ sufficiently large):

**Definition 3.1.** [Simple Multi-View Setting] For each class $y \in [k]$, let $v_{y,1}, v_{y,2} \in \mathbb{R}^d$ be orthonormal unit vectors also satisfying $v_{y,\ell} \perp v_{s,\ell'}$ when $y \neq s$ for any $\ell, \ell' \in [2]$. Each point $(x, y) \sim \mathcal{D}$ is then generated by sampling $y \in [k]$ uniformly and constructing $x$ as:

$$x = \beta_y v_{y,1} + (1 - \beta_y) v_{y,2} \qquad \beta_y \sim \mathrm{Uni}([0.1, 0.9]) \tag{3.2}$$

Definition 3.1 is **multi-view** in the following sense: for any class $y$, it suffices (from an accuracy perspective) to learn a model $g$ that has a significant correlation with *either* the feature vector $v_{y,1}$ or $v_{y,2}$. In this context, one can think of feature learning as corresponding to how positively correlated the weight $w_y$ is with each of the same class feature vectors $v_{y,1}$ and $v_{y,1}$ (we provide a more rigorous definition in our main results).

If one now considers the empirical cross-entropy loss $J(g, \mathcal{X})$, it is straightforward to see that it is possible to achieve the global minimum of $J(g, \mathcal{X})$ by just considering models $g$ in which we take $\langle w_y, v_{y,1} \rangle \to \infty$ for every class $y$. This means we can minimize the usual cross-entropy loss without learning both features in the dataset $\mathcal{X}$.

However, this is *not* the case for Midpoint Mixup. Indeed, we show below that a necessary (with extremely high probability) and sufficient condition for a linear model $g$ to minimize $J_{MM}$ (when

taking its scaling to $\infty$) is that it has equal correlation with both features for every class (sufficiency relies also on having weaker correlations with other class features). In what follows, we use $\inf J_{MM}(h, \mathcal{X})$ to indicate the global minimum of $J_{MM}$ over *all* functions $h : \mathbb{R}^d \to \mathbb{R}^k$ (i.e. this is the smallest achievable loss).

**Lemma 3.2.** [Midpoint Mixup Optimal Direction] A linear model $g$ satisfies the following:

$$\lim_{\gamma \to \infty} J_{MM}(\gamma g, \mathcal{X}) = \inf J_{MM}(h, \mathcal{X}) \tag{3.3}$$

If $g$ has the property that for every class $y$ we have $\langle w_y, v_{y,\ell_1} \rangle = \langle w_s, v_{s,\ell_2} \rangle > 0$ and $\langle w_y, v_{s,\ell_2} \rangle \leq 0$ for every $s \neq y$ and $\ell_1, \ell_2 \in [2]$. Furthermore, with probability $1 - \exp(-\Theta(N))$ (over the randomness of $\mathcal{X}$), the condition $\langle w_y, v_{y,\ell_1} \rangle = \langle w_s, v_{s,\ell_2} \rangle$ is necessary for $g$ to satisfy Equation 3.3.

**Proof Sketch.** The idea is that if $g$ has equal correlation with both features for every class, its predictions will be constant on the original data points due to the fact that the coefficients for each feature in each data point are mirrored as per Equation 3.2. With the condition $\langle w_y, v_{s,\ell} \rangle \leq 0$ (this can be weakened significantly), this implies the softmax output of $g$ on the Midpoint Mixup points will be exactly $1/2$ for each of the classes being mixed (and 0 for all other classes), which is optimal.

As mentioned earlier, we can also show that if we consider $J_M(g, \mathcal{X}, \mathcal{D}_\lambda)$ for any other non-point-mass distribution, we can prove that the analogue of Lemma 3.2 does not hold true.

**Proposition 3.3.** For any distribution $\mathcal{D}_\lambda$ that is not a point mass on $0, 1$, or $1/2$, and any linear model $g$ satisfying the conditions of Lemma 3.2, we have that with probability $1 - \exp(-\Theta(N))$ (over the randomness of $\mathcal{X}$) there exists an $\epsilon_0 > 0$ depending only on $\mathcal{D}_\lambda$ such that:

$$J_M(g, \mathcal{X}, \mathcal{D}_\lambda) \geq \inf J_M(h, \mathcal{X}, \mathcal{D}_\lambda) + \epsilon_0 \tag{3.4}$$

**Proof Sketch.** In the case of general mixing distributions, we cannot achieve the Mixup optimal behavior of $\phi^{y_i}(g(z_{i,j}(\lambda))) = \lambda$ for every $\lambda$ if the outputs $g^y$ are constant on the original data points.

Lemma 3.2 outlines the key theoretical benefit of Midpoint Mixup - namely that its global optimizers exist within the class of models that we consider, and such optimizers learn all features in the data equally. And although Lemma 3.2 is stated in the context of linear models, the result naturally carries through to when we consider two-layer neural networks of the type we define in the next section. That being said, the interpretation of Proposition 3.3 is not intended to disqualify the possibility that the minimizer of $J_M(g, \mathcal{X}, \mathcal{D}_\lambda)$ when restricted to a specific model class is a model in which all features are learned near-equally (we expect this to be the case in fact for any reasonable $\mathcal{D}_\lambda$). Proposition 3.3 is moreso intended to motivate the study of Midpoint Mixup as a particularly interesting choice of the mixing distribution $\mathcal{D}_\lambda$.

We now proceed one step further from the above results and show that the feature learning benefit of Midpoint Mixup manifests itself even in the optimization process (when using gradient-based methods). We show that, if significant separation between feature correlations exists, the Midpoint Mixup gradients correct the separation. For simplicity, we suppose WLOG that $\langle w_y, v_{y,1} \rangle > \langle w_y, v_{y,2} \rangle$. Now letting $\Delta_y = \langle w_y, v_{y,1} - v_{y,2} \rangle$ and using the notation $\nabla_{w_y}$ for $\frac{\partial}{\partial w_y}$, we can prove:

**Proposition 3.4.** [Mixup Gradient Lower Bound] Let $y$ be any class such that $\Delta_y \geq \log k$, and suppose that both $\langle w_y, v_{y,1} \rangle \geq 0$ and the cross-class orthogonality condition $\langle w_s, v_{u,\ell} \rangle = 0$ hold for all $s \neq u$ and $\ell \in [2]$. Then we have with high probability that:

$$\langle -\nabla_{w_y} J_{MM}(g, \mathcal{X}), v_{y,2} \rangle \geq \Theta\left(\frac{1}{k^2}\right) \tag{3.5}$$

**Proof Sketch.** The key idea is to analyze the gradient correlation with the direction $v_{y,1} - v_{y,2}$ via a concentration of measure argument.

Proposition 3.4 shows that, assuming nonnegativity of within-class correlations and an orthogonality condition across classes (which we will show to be approximately true in our main results), the feature correlation that is lagging behind for any class $y$ will receive a significant gradient when optimizing the Midpoint Mixup loss. On the other hand, we can also prove that this need not be the case for empirical risk minimization:

**Proposition 3.5.** [ERM Gradient Upper Bound] For every $y \in [k]$, assuming the same conditions as in Proposition 3.4, if $\Delta_y \geq C \log k$ for any $C > 0$ then with high probability we have that:

$$\langle -\boldsymbol{\nabla}_{w_y} J(g, \mathcal{X}), v_{y,2} \rangle \leq O\left(\frac{1}{k^{0.1C-1}}\right) \tag{3.6}$$

**Proof Sketch.** This follows directly from the form of the gradient for $J(g, \mathcal{X})$.

While Proposition 3.5 demonstrates that training using ERM can possibly fail to learn both features associated with a class due to increasingly small gradients, one can verify that this does not naturally occur in the optimization dynamics of linear models on linearly separable data of the type in Definition 3.1 (see for example, the related result in Chidambaram et al. (2021)). On the other hand, if we move away from linearly separable data and linear models to more realistic settings, the situation described above does indeed show up, which motivates our main results.

## 4 ANALYZING MIDPOINT MIXUP TRAINING DYNAMICS ON GENERAL MULTI-VIEW DATA

For our main results, we now consider a data distribution and class of models that are meant to more closely mimic practical situations.

### 4.1 GENERAL MULTI-VIEW DATA SETUP

We adopt a slightly simplified version of the setting of Allen-Zhu and Li (2021). We still consider the problem of $k$-class classification on a dataset $\mathcal{X}$ of $N$ labeled data points, but our data points are now represented as ordered tuples $x = (x^{(1)}, ..., x^{(P)})$ of $P$ input patches $x^{(i)}$ with each $x^{(i)} \in \mathbb{R}^d$ (so $\mathcal{X} \subset \mathbb{R}^{Pd} \times [k]$).

As was the case in Definition 3.1 and in Allen-Zhu and Li (2021), we assume that the data is multi-view in that each class $y$ is associated with 2 orthonormal feature vectors $v_{y,1}$ and $v_{y,2}$, and we once again consider $N$ and $k$ to be sufficiently large. As mentioned in Allen-Zhu and Li (2021), we could alternatively consider the number of classes $k$ to be fixed (i.e. binary classification) and the number of associated features to be large, and our theory would still translate. We now precisely define the data generating distribution $\mathcal{D}$ that we will focus on for the remainder of the paper.

**Definition 4.1.** [General Multi-View Data Distribution] Identically to Definition 3.1, each class $y$ is associated with two orthonormal feature vectors, after which each point $(x, y) \sim \mathcal{D}$ is generated as:.

1. Sample a label $y$ uniformly from $[k]$.

2. Designate via any method two disjoint subsets $P_{y,1}(x), P_{y,2}(x) \subset [P]$ with $|P_{y,1}(x)| = |P_{y,2}(x)| = C_P$ for a universal constant $C_P$, and additionally choose via any method a bijection $\varphi : P_{y,1}(x) \to P_{y,2}(x)$. We then generate the **signal patches** of $x$ in corresponding pairs $x^{(p)} = \beta_{y,p} v_{y,1}$ and $x^{(\varphi(p))} = (\delta_2 - \beta_{y,p}) v_{y,2} = \beta_{y,\varphi(p)} v_{y,2}$ for every $p \in P_{y,1}(x)$ with the $\beta_{y,p}$ chosen according to a symmetric distribution (allowed to vary per class $y$) supported on $[\delta_1, \delta_2 - \delta_1]$ satisfying the anti-concentration property that $\beta_{y,p}$ takes values in a subset of its support whose Lebesgue measure is $O(1/\log k)$ with probability $o(1)$.[1]

3. Fix, via any method, $Q$ distinct classes $s_1, s_2, ..., s_Q \in [k] \setminus y$ with $Q = \Theta(1)$. The remaining $[P] \setminus (P_{y,1}(x) \cup P_{y,2}(x))$ patches not considered above are the **feature noise patches** of $x$, and are defined to be $x^{(p)} = \sum_{j \in [Q]} \sum_{\ell \in [2]} \gamma_{j,\ell} v_{s_j,\ell}$, where the $\gamma_{j,\ell} \in [\delta_3, \delta_4]$ can be arbitrary.

Note that there are parts of the data-generating process that we leave underspecified, as our results will work for any choice. Henceforth, we use $\mathcal{X}$ to refer to a dataset consisting of $N$ i.i.d. draws from the distribution $\mathcal{D}$. Our data distribution represents a very low signal-to-noise (SNR) setting in which the true signal for a class exists only in a constant ($2C_P$) number of patches while the rest of the patches contain low magnitude noise in the form of other class features.

---

[1]This assumption is true for any distribution with reasonable variance; for example, the uniform distribution.

We focus on the case of learning the data distribution $\mathcal{D}$ with the same two-layer CNN-like architecture used in Allen-Zhu and Li (2021). We recall that this architecture relies on the following polynomially-smoothed ReLU activation, which we refer to as $\widetilde{\text{ReLU}}$:

$$
\widetilde{\text{ReLU}}(x) = \begin{cases} 0 & \text{if } x \leq 0 \\ \frac{x^\alpha}{\alpha \rho^{\alpha-1}} & \text{if } x \in [0, \rho] \\ x - \left(1 - \frac{1}{\alpha}\right)\rho & \text{if } x \geq \rho \end{cases}
$$

The polynomial part of this activation function will be very useful for us in suppressing the feature noise in $\mathcal{D}$. Our full network architecture, which consists of $m$ hidden neurons, can then be specified as follows.

**Definition 4.2.** [2-Layer Network] We denote our network by $g : \mathbb{R}^{Pd} \to \mathbb{R}^k$. For each $y \in [k]$, we define $g^y$ as follows.

$$
g^y(x) = \sum_{r \in [m]} \sum_{p \in [P]} \widetilde{\text{ReLU}}\left(\left\langle w_{y,r}, x^{(p)} \right\rangle\right) \tag{4.1}
$$

We will use $w_{y,r}^{(0)}$ to refer to the weights of the network $g$ at initialization (and $w_{y,r}^{(t)}$ after $t$ steps of gradient descent), and similarly $g_t$ to refer to the model after $t$ iterations of gradient descent. We consider the standard choice of Xavier initialization, which, in our setting, corresponds to $w_{y,r}^{(0)} \sim \mathcal{N}(0, \frac{1}{d}\mathbf{I}_d)$.

For model training, we focus on full batch gradient descent with a fixed learning rate of $\eta$ applied to $J(g, \mathcal{X})$ and $J_{MM}(g, \mathcal{X})$. Once again using the notation $\boldsymbol{\nabla}_{w_{y,r}^{(t)}}$ for $\frac{\partial}{\partial w_{y,r}^{(t)}}$, the updates to the weights of the network $g$ are thus of the form:

$$
w_{y,r}^{(t+1)} = w_{y,r}^{(t)} - \eta \boldsymbol{\nabla}_{w_{y,r}^{(t)}} J_{MM}(g, \mathcal{X}) \tag{4.2}
$$

In defining our data distribution and model above, we have introduced several hyperparameters. Throughout our results, we make the following assumptions about these hyperparameters.

**Assumption 4.3.** [Choice of Hyperparameters] We assume that:

$$
\begin{array}{cccc} d = \Omega(k^{20}) & P = \Theta(k^2) & C_P = \Theta(1) & m = \Theta(k) \\ \delta_1, \delta_2 = \Theta(1) & \delta_3, \delta_4 = \Theta(k^{-1.5}) & \rho = \Theta(1/k) & \alpha = 8 \end{array}
$$

**Discussion of Hyperparameter Choices.** We make concrete choices of hyperparameters above for the sake of calculations (and we stress that these are not close to the tightest possible choices), but only the relationships between them are important. Namely, we need $d$ to be a significantly larger polynomial of $k$ than $P$, we need $\delta_3, \delta_4 = o(1)$ but large enough so that $P\delta_3 \gg \delta_2$ (to avoid learnability by linear models, as shown below), we need $\alpha$ sufficiently large so that the network can suppress the low-magnitude feature noise, and we need $\delta_1, \delta_2 = \Theta(1)$ so that the signal feature coefficients significantly outweigh the noise feature coefficients.

To convince the reader that our choice of model is not needlessly complicated given the setting, we prove the following result showing that there exist realizations of the distribution $\mathcal{D}$ on which linear classifiers cannot achieve perfect accuracy.

**Proposition 4.4.** There exists a $\mathcal{D}$ satisfying all of the conditions of Definition 4.1 and Assumption 4.3 such that with probability at least $1 - k^2 \exp(\Theta(-N/k^2))$, for any classifier $h : \mathbb{R}^{Pd} \to \mathbb{R}^k$ of the form $h^y(x) = \sum_{p \in [P]} \left\langle w_y, x^{(p)} \right\rangle$ and any $\mathcal{X}$ consisting of $N$ i.i.d. draws from $\mathcal{D}$, there exists a point $(x, y) \in \mathcal{X}$ and a class $s \neq y$ such that $h^s(x) \geq h^y(x)$.

**Proof Sketch.** The idea, as was originally pointed out by Allen-Zhu and Li (2021), is that there are $\Theta(k^2)$ feature noise patches with coefficients of order $\Theta(k^{-1.5})$. Thus, because the features are orthogonal, these noise patches can influence the classification by an order $\Theta(\sqrt{k})$ term away from the direction of the true signal.

The full proof can be found in Section C of the Appendix.

### 4.2 MAIN RESULTS

Having established the setting for our main results, we now concretely define the notion of feature learning in our context.

**Definition 4.5.** [Feature Learning] Let $(x, y) \sim \mathcal{D}$. We say that feature $v_{y,\ell}$ is *learned* by $g$ if $\operatorname{argmax}_s g^s(x') = y$ where $x'$ is $x$ with all instances of feature $v_{y,3-\ell}$ replaced by the all-zero vector.

Our definition of feature learning corresponds to whether the model $g$ is able to correctly classify data points in the presence of only a single signal feature instead of both (generalizing the notion of weight-feature correlation to nonlinear models). By analyzing the gradient descent dynamics of $g$ for the empirical cross-entropy $J$, we can then show the following.

**Theorem 4.6.** For $k$ and $N$ sufficiently large and the settings stated in Assumption 4.3, we have that the following hold with probability at least $1 - O(1/k)$ after running gradient descent with a step size $\eta = O(1/\operatorname{poly}(k))$ for $O(\operatorname{poly}(k)/\eta)$ iterations on $J(g, \mathcal{X})$ (for sufficiently large polynomials in $k$):

1. (Training accuracy is perfect): For all $(x_i, y_i) \in \mathcal{X}$, we have $\operatorname{argmax}_s g_t^s(x_i) = y_i$.

2. (Only one feature is learned): For $(1 - o(1))k$ classes, there exists exactly one feature that is learned in the sense of Definition 4.5 by the model $g_t$.

Furthermore, the above remains true for all $t = O(\operatorname{poly}(k))$ for any polynomial in $k$.

**Proof Sketch.** The proof is in spirit very similar to Theorem 1 in Allen-Zhu and Li (2021), and relies on many of the tools therein. The main idea is that, with high probability, there exists a separation between the class $y$ weight correlations with the features $v_{y,1}$ and $v_{y,2}$ at initialization. This separation is then amplified throughout training due to the polynomial part of $\widetilde{\operatorname{ReLU}}$. Once one feature correlation becomes large enough, the gradient updates to the class $y$ weights rapidly decrease, leading to the remaining feature not being learned.

Theorem 4.6 shows that only one feature is learned (in our sense) for the vast majority of classes. As mentioned, our proof is quite similar to Allen-Zhu and Li (2021), but due to simplifications in our setting (no added Gaussian noise for example) and some different ideas the proof is much shorter - we hope this makes some of the machinery from Allen-Zhu and Li (2021) accessible to a wider audience.

The reason we prove Theorem 4.6 is in fact to highlight the contrast provided by the analogous result for Midpoint Mixup.

**Theorem 4.7.** For $k$ and $N$ sufficiently large and the settings stated in Assumptions 4.3, we have that the following hold with probability $1 - O(1/k)$ after running gradient descent with a step size $\eta = O(1/\operatorname{poly}(k))$ for $O(\operatorname{poly}(k)/\eta)$ iterations on $J_{MM}(g, \mathcal{X})$ (for sufficiently large polynomials in $k$):

1. (Training accuracy is perfect): For all $(x_i, y_i) \in \mathcal{X}$, we have $\operatorname{argmax}_s g^s(x_i) = y_i$.

2. (Both features are learned): For each class $y \in [k]$, both $v_{y,1}$ and $v_{y,2}$ are learned in the sense of Definition 4.5 by the model $g$.

Furthermore, the above remains true for all $t = O(\operatorname{poly}(k))$ for any polynomial in $k$.

**Proof Sketch.** The core idea of the proof relies on similar techniques to that of Proposition 3.4, but the nonlinear part of the $\widetilde{\operatorname{ReLU}}$ activation introduces a few additional difficulties due to the fact that the gradients in the nonlinear par are much smaller than those in the linear part of $\widetilde{\operatorname{ReLU}}$. Nevertheless, we show that even these smaller gradients are sufficient for the feature correlation that is lagging behind to catch up in polynomial time.

The full proofs of Theorems 4.6 and 4.7 can be found in Section B of the Appendix.

**Remark 4.8.** Theorems 4.6 and 4.7 show a separation between ERM and Midpoint Mixup with respect to feature learning, as we have defined. They are *not* results regarding the test accuracy of the trained models on the distribution $\mathcal{D}$; even learning only a single feature per class is sufficient for perfect test accuracy on $\mathcal{D}$. The significance (and our desired interpretation) of these results is that,

when the training distribution $\mathcal{D}$ has some additional spurious features when compared to the testing distribution, ERM can potentially fail to learn the true signal features whereas Midpoint Mixup will likely learn all features (including the true signal). One may also interpret the results as generalization that is robust to distributional shift; the test distribution in this case has dropped some features present in the training distribution.

## 5 EXPERIMENTS

The goal of the results of Sections 3 and 4 was to provide theory (from a feature learning and optimization perspective) for why Mixup has enjoyed success over ERM in many practical settings. The intuition is that, for image classification tasks, one could reasonably expect images from the same class to be generated from a shared set of latent features (much like our data distribution in Definition 4.1), in which case it may be possible to achieve perfect training accuracy by learning a strict subset of these features when doing empirical risk minimization. On the other hand, based on our ideas, we would expect Mixup to learn all such latent features associated with each class (assuming some dependency between them), and thus potentially generalize better.

A direct empirical verification of this phenomenon on image datasets is tricky (and a possible avenue for future work) due to the fact that one would need to clearly define a notion of latent features with respect to the images being considered, which is outside the scope of this work. Instead, we take for granted that such features *exist*, and attempt to verify whether Mixup is able to learn the "true" features associated with each class better than ERM when spurious features are added.

For our experimental setup, we consider training ResNet-18(He et al., 2015) on versions of Fashion MNIST (FMNIST) (Xiao et al., 2017), CIFAR-10, and CIFAR-100 (Krizhevsky, 2009) in which every training data point is transformed such that a randomly sampled training point from *a different* (but randomly fixed) class is concatenated (along the channels dimension) to the original point. Additionally, to introduce a dependency structure akin to what we have in Definitions 3.1 and 4.1, we sample a $\gamma \sim \mathrm{Uni}([0, 1])$ and scale the first part of the training point (the true image) by $\gamma$ while scaling the concatenated part by $1 - \gamma$ during training.

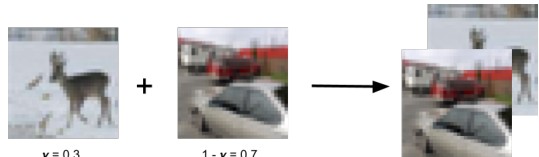

Figure 1: Visualization of data modification in CIFAR-10.

If we work under the intuition that images from each class are generated by relatively different latent features, then this modification process corresponds to adding patches of (fixed) spurious features to each class that have a dependency (from the scaling factor $\gamma$) on the original features of the data. We leave the test data for each dataset **unmodified**, except for the concatenation of an all-zeros vector of the same shape to each point so that the shape of the test data matches that of the training data (in effect, this penalizes models that learned only the spurious features we concatenated in the training data). This zeroing out of the additional channels is also intended to replicate Definition 4.5 in our experimental setup.

While we consider the above setup to be intuitive and resemble our theoretical setting, it is fair to ask why we chose this setup compared to the many possible alternatives. Firstly, we found that using synthetic spurious features (i.e. random orthogonal vectors scaled to have the same norm as the images) as opposed to images from different classes was far too noisy (training error went to 0 immediately); the test errors on each dataset degraded to near-random levels, so it was difficult to make comparisons. Additionally, we found the same to be true if we considered adding spurious features as opposed to concatenating them.

For each of our image classification tasks, we train models using Mixup with $\mathcal{D}_\lambda = \mathrm{Beta}(1, 1)$ (the choice used in Zhang et al. (2018) for CIFAR, which we refer to as Uniform Mixup), Midpoint Mixup, and ERM. Our implementation is in PyTorch (Paszke et al., 2019) and uses the ResNet

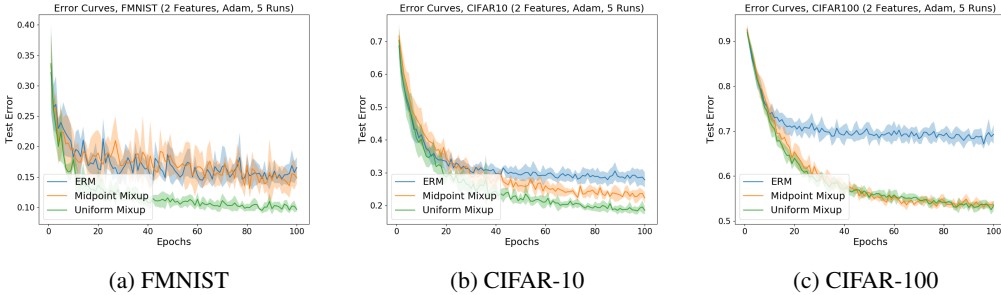

(a) FMNIST  (b) CIFAR-10  (c) CIFAR-100

Figure 2: Test error comparison between Uniform Mixup (green), Midpoint Mixup (orange), and ERM (blue). Each curve represents the average of 5 model runs (over the randomness of the data augmentations and model initializations), while the surrounding area represents 1 standard deviation.

implementation of Kuang Liu, released under an MIT license. All models were trained for 100 epochs with a batch size of 750, which was the largest feasible size on our compute setup of a single P100 GPU (we use a large batch size to approximate the full batch gradient descent aspect of our theory). For optimization, we use Adam (Kingma and Ba, 2015) with the default hyperparameters of $\beta_1 = 0.9, \beta_2 = 0.999$ and a learning rate of $0.001$. We did a modest amount of hyperparameter tuning in preliminary experiments, where we compared Adam and SGD with different log-spaced learning rates in the range $[0.001, 0.1]$, and found that Adam with the default hyperparameters almost always worked the best. We report our results for each dataset in Table 1, and accompanying test error plots are shown in Figure 2.

| Model | FMNIST | CIFAR-10 | CIFAR-100 |
|---|---|---|---|
| Uniform Mixup | **9.66** | **18.52** $\pm 1$ | **53.42** $\pm 1$ |
| Midpoint Mixup | 14.84 $\pm 1$ | 22.29 $\pm 2$ | 53.61 $\pm 2$ |
| ERM | 16.55 $\pm 2$ | 27.77 $\pm 2$ | 69.28 $\pm 2$ |

Table 1: Final test errors on *unmodified test data* (mean over 5 runs) along with 1 standard deviation range for Uniform Mixup, Midpoint Mixup, and ERM.

From Table 1 we see that Uniform Mixup performs the best in all cases, and that Midpoint Mixup tracks the performance of Uniform Mixup reasonably closely. We stress that the ordering of model performance is unsurprising; a truly fair comparison with Midpoint Mixup would require training on all $N^2$ possible mixed points, which is infeasible in our compute setup (we opt to randomly mix points per batch, as is standard). Our experiments are intended to show that Midpoint Mixup still non-trivially captures the benefits of Mixup in an empirical setting that is far from the asymptotic regime of our theory, while Mixup using standard hyperparameter settings significantly outperforms ERM in the presence of spurious features. A final observation worth making is that we find Midpoint Mixup performs significantly better than ERM when moving from the 10-class settings of FMNIST and CIFAR-10 to the 100-class setting of CIFAR-100, and this is in line with what our theory predicts (a larger number of classes more closely approximates our setting).

## 6   CONCLUSION

To summarize, the main contributions of this work have been theoretical motivation for an extreme case of Mixup training (Midpoint Mixup), as well as an optimization analysis separating the learning dynamics of a 2-layer convolutional network trained using Midpoint Mixup and empirical risk minimization.

Our results show that, for a class of data distributions satisfying the property that there are multiple, dependent features correlated with each class in the data, Midpoint Mixup can outperform ERM (both theoretically and empirically) in learning these features. We hope that the ideas introduced in the theory can be a useful building block for future theoretical investigations into Mixup and related methods in the context of training neural networks.

## 7 ETHICS STATEMENT

As this work is almost entirely theoretical in nature, we do not anticipate any (direct) negative broader impacts or potential for misuse.

## 8 REPRODUCIBILITY STATEMENT

All of the results discussed in this paper have accompanying complete proofs in Sections B and C of the Appendix. While the proofs are quite technical in nature, we have tried our best to provide intuitive explanations at each step, and have also included derivations of various calculations and well-known concentration of measure results in Section A of the Appendix.

We have also included in the supplementary material the code necessary to run the experiments in Section 5, along with detailed instructions explaining how to recreate each of our experiments.

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

# A  SUPPORTING LEMMAS AND CALCULATIONS

In this section we collect several technical lemmas and computations that will be necessary for the proofs of our main results.

## A.1  GAUSSIAN CONCENTRATION AND ANTI-CONCENTRATION RESULTS

The following are well-known concentration results for Gaussian random variables; we include proofs for the convenience of the reader.

**Proposition A.1.** Let $X_i \sim \mathcal{N}(0, \sigma_i^2)$ with $i \in [m]$ and let $\sigma = \max_i \sigma_i$. Then,

$$\mathbb{E}[\max_i X_i] \leq \sigma\sqrt{2\log m}$$

*Proof.* Let $Z = \max_i X_i$. Then by Jensen's inequality and the MGF of $\mathcal{N}(0, \sigma_i^2)$, we have:

$$\exp\left(t\mathbb{E}[Z]\right) \leq \mathbb{E}\exp(tZ) = \mathbb{E}[\exp\left(t\max_i X_i\right)]$$

$$\leq \mathbb{E}\big[\sum_i \exp(tX_i)\big] = \sum_i \exp\left(t^2\sigma_i^2/2\right)$$

$$\leq m\exp\left(t^2\sigma^2/2\right)$$

$$\implies \mathbb{E}[Z] \leq \frac{\log m}{t} + \frac{t\sigma^2}{2}$$

Minimizing the RHS yields $t = \sqrt{2\log m}/\sigma$, from which the result follows. $\qquad\square$

**Proposition A.2.** Let $X_i$ be as in Proposition A.1. Then,

$$P(\max_i X_i \geq t + \sigma\sqrt{2\log m}) \leq \exp\left(-t^2/(2\sigma^2)\right)$$

*Proof.* We simply union bound and use the fact that $P(X_i \geq t) \leq \exp\left(-t^2/(2\sigma^2)\right)$ (Chernoff bound for zero mean Gaussians) to get:

$$P(\max_i X_i \geq t + \sigma\sqrt{2\log m}) \leq \sum_i P(X_i \geq t + \sigma\sqrt{2\log m})$$

$$\leq m\exp\left(-(t + \sigma\sqrt{2\log m})^2/(2\sigma^2)\right)$$

$$\leq \exp\left(-t^2/(2\sigma^2)\right)$$

$$\square$$

**Proposition A.3.** Let $X_1, X_2, ..., X_m$ be i.i.d. Gaussian variables with mean 0 and variance $\sigma^2$. Then we have that:

$$P\left(\max_i X_i > \Theta\left(\sigma\sqrt{\log(m/\log(1/\delta))}\right)\right) = 1 - \Theta(\delta)$$

*Proof.* We recall that:

$$P(X_i > x) = \Theta\left(\frac{\sigma}{x}e^{-x^2/(2\sigma^2)}\right)$$

A proof of this fact can be found in Vershynin (2018). We additionally have that:

$$P(\max X_i > x) = 1 - (1 - P(X_i > x))^m$$

So from the previous asymptotic characterization of $P(X_i > x)$ we have that choosing $x = \Theta\left(\sigma\sqrt{\log(m/\log(1/\delta))}\right)$ gives $P(X_i > x) = \Theta(\log(1/\delta)/m)$, from which the result follows. $\quad\square$

We will also have need for a recent anti-concentration result due to Chernozhukov et al. (2014), which we restate below.

**Proposition A.4** (Theorem 3 (i) Chernozhukov et al. (2014)). Let $X_i \sim \mathcal{N}(0, \sigma^2)$ for $i \in [m]$ with $\sigma^2 > 0$. Defining $a_m = \mathbb{E}[\max_i X_i / \sigma]$, we then have for every $\epsilon > 0$:

$$\sup_{x \in \mathbb{R}} P\left(\left|\max_i X_i - x\right| \leq \epsilon\right) \leq 4\epsilon(1 + a_m)/\sigma$$

**Corollary A.5.** Applying Proposition A.1, we have $\sup_{x \in \mathbb{R}} P\left(|\max_i X_i - x| \leq \epsilon\right) \leq 4\epsilon(1 + \sqrt{2\log m})/\sigma$.

### A.2 GRADIENT CALCULATIONS

Here we collect the gradient calculations used in the proofs of the main results. We recall that we use $\mathbf{\nabla}_{w_{y,r}^{(t)}}$ to indicate $\frac{\partial}{\partial w_{y,r}^{(t)}}$ and $z_{i,j} = (x_i + x_j)/2$. Additionally, we will omit parentheses after $\widetilde{\text{ReLU}}$ when function application is clear.

**Calculation A.6.** For any $(x_i, y_i) \in \mathcal{X}$:

$$\mathbf{\nabla}_{w_{y_i,r}^{(t)}} g_t^{y_i}(x_i) = \sum_{p \in [P]} \widetilde{\text{ReLU}}' \left\langle w_{y_i,r}, x_i^{(p)} \right\rangle x_i^{(p)}$$

*Proof.*

$$\mathbf{\nabla}_{w_{y_i,r}^{(t)}} g_t^y(x_i) = \frac{\partial}{\partial w_{y_i,r}^{(t)}} \sum_{u \in [m]} \sum_{p \in [P]} \widetilde{\text{ReLU}} \left\langle w_{y_i,u}^{(t)}, x_i^{(p)} \right\rangle = \sum_{p \in [P]} \widetilde{\text{ReLU}}' \left\langle w_{y_i,r}^{(t)}, x_i^{(p)} \right\rangle x_i^{(p)}$$

$\square$

**Calculation A.7.** For any $(x_i, y_i) \in \mathcal{X}$, if $\max_{u \in [k]} \left\langle w_{y_i,r}^{(t)}, v_{u,\ell} \right\rangle < \rho/(\delta_2 - \delta_1)$ and $s \neq y_i$, then:

$$\left\langle \mathbf{\nabla}_{w_{y_i,r}^{(t)}} g_t^{y_i}(x_i), v_{y_i,\ell} \right\rangle = \sum_{p \in P_{y_i,\ell}(x_i)} \frac{\beta_{i,p}^\alpha \left\langle w_{y_i,r}^{(t)}, v_{y_i,\ell} \right\rangle^{\alpha-1}}{\rho^{\alpha-1}}$$

$$\left\langle \mathbf{\nabla}_{w_{y_i,r}^{(t)}} g_t^{y_i}(x_i), v_{s,\ell} \right\rangle \leq \Theta\left(\frac{P\delta_4^\alpha \max_{u \neq y} \left\langle w_{y_i,r}^{(t)}, v_{u,\ell} \right\rangle^{\alpha-1}}{\rho^{\alpha-1}}\right)$$

*Proof.* When $\max_{u \in [k]} \left\langle w_{y_i,r}^{(t)}, v_{u,\ell} \right\rangle < \rho/(\delta_2 - \delta_1)$, we are in the polynomial part of $\widetilde{\text{ReLU}}$ for every patch in $x_i$, since $\max_{p \in P_{y_i,\ell}(x_i)} \left\langle w_{y_i,r}^{(t)}, x_i^{(p)} \right\rangle < \rho$ since $\beta_{i,p} \leq \delta_2 - \delta_1$. The first line then follows from Calculation A.6 and the fact that all of the feature vectors are orthonormal (so only those patches that have the features $v_{y_i,\ell}$ are relevant). The second line follows from the fact that there are at most $P - 2C_P$ feature noise patches containing the vector $v_{s,\ell}$, and in each of these patches there are only a constant number of feature vectors (which we do not constrain). $\square$

**Calculation A.8.** For any $(x_i, y_i) \in \mathcal{X}$, if $\left\langle w_{y_i,r}^{(t)}, v_{y_i,\ell} \right\rangle \geq \rho/\delta_1$, then:

$$\left\langle \mathbf{\nabla}_{w_{y_i,r}^{(t)}} g_t^{y_i}(x_i), v_{y_i,\ell} \right\rangle = \sum_{p \in P_{y_i,\ell}(x_i)} \beta_{i,p}$$

*Proof.* When $\left\langle w_{y_i,r}^{(t)}, v_{y_i,\ell} \right\rangle \geq \rho/\delta_1$ we necessarily have $\min_{p \in P_{y_i,\ell}(x_i)} \left\langle w_{y_i,r}^{(t)}, x_i^{(p)} \right\rangle \geq \rho$ since $\beta_{i,p} \geq \delta_1$, and then the result again follows from Calculation A.6 and the fact that $\widetilde{\text{ReLU}}' = 1$ in the linear regime. $\square$

**Calculation A.9** (ERM Gradient).

$$\mathbf{\nabla}_{w_{y,r}^{(t)}} J(g_t, \mathcal{X}) = -\frac{1}{N} \sum_{i \in [N]} \left(\mathbb{1}_{y_i = y} - \phi^y\big(g(x_i)\big)\right) \mathbf{\nabla}_{w_{y,r}^{(t)}} g_t^y(x_i)$$

*Proof.* First let us observe that:

$$\log \phi^{y_i}(g_t(x_i)) = g_t^{y_i}(x_i) - \log \sum_s \exp(g_t^s(x_i))$$

$$\implies \frac{\partial \log \phi^{y_i}(g_t(x_i))}{\partial w_{y,r}} = \mathbb{1}_{y_i=y} \nabla_{w_{y,r}^{(t)}} g_t^y(x_i) - \phi^y(g(x_i)) \nabla_{w_{y,r}^{(t)}} g_t^y(x_i)$$

Summing (and negating) the above over all points $x_i$ gives the result. $\qquad\square$

**Calculation A.10** (Midpoint Mixup Gradient)**.**

$$\nabla_{w_{y,r}^{(t)}} J_{MM}(g_t, \mathcal{X}) = -\frac{1}{2N^2} \sum_{i \in [N]} \sum_{j \in [N]} \left( \mathbb{1}_{y_i=y} + \mathbb{1}_{y_j=y} - 2\phi^y\big(g_t(z_{i,j})\big) \right) \nabla_{w_{y,r}^{(t)}} g_t^y(z_{i,j})$$

*Proof.* Follows from applying Calculation A.9 to each part of the summation in $J_{MM}(g, \mathcal{X})$. $\qquad\square$

## B  PROOFS OF MAIN RESULTS

This section contains the proofs of the main results in this paper. We have opted to present the proofs in a linear fashion - inlining several claims and their proofs along the way - as we find this to be more readable than the alternative. The proofs of inlined claims are ended with the ■ symbol, while the proofs of the overarching results are ended with the □ symbol.

For convenience, we recall the assumptions (as they were stated in the main body) that are used in these results:

**Assumption 4.3.** [Choice of Hyperparameters] We assume that:

$$d = \Omega(k^{20}) \qquad P = \Theta(k^2) \qquad C_P = \Theta(1) \qquad m = \Theta(k)$$

$$\delta_1, \delta_2 = \Theta(1) \qquad \delta_3, \delta_4 = \Theta(k^{-1.5}) \qquad \rho = \Theta(1/k) \qquad \alpha = 8$$

### B.1  PROOF OF THEOREM 4.6

**Theorem 4.6.** For $k$ and $N$ sufficiently large and the settings stated in Assumption 4.3, we have that the following hold with probability at least $1 - O(1/k)$ after running gradient descent with a step size $\eta = O(1/\mathrm{poly}(k))$ for $O(\mathrm{poly}(k)/\eta)$ iterations on $J(g, \mathcal{X})$ (for sufficiently large polynomials in $k$):

1. (Training accuracy is perfect): For all $(x_i, y_i) \in \mathcal{X}$, we have $\operatorname{argmax}_s g_t^s(x_i) = y_i$.

2. (Only one feature is learned): For $(1 - o(1))k$ classes, there exists exactly one feature that is learned in the sense of Definition 4.5 by the model $g_t$.

Furthermore, the above remains true for all $t = O(\mathrm{poly}(k))$ for any polynomial in $k$.

*Proof.* We break the proof into two parts. In part one, we prove that (with high probability) each class output $g_t^y$ becomes large (but not too large) on data points belonging to class $y$ and stays small on other data points, which consequently allows us to obtain perfect training accuracy at the end (thereby proving the first half of the theorem). In part two, we show that (again with high probability) the max correlations with features $v_{y,1}$ and $v_{y,2}$ for a class $y$ have a separation at initialization that gets amplified over the course of training, and due to this separation one of the feature correlations becomes essentially irrelevant, which will be used to prove the second half of the theorem.

**Part I.**

In this part, we show that the network output $g_t^{y_i}(x_i)$ reaches and remains $\Theta(\log k)$ while $g_t^s(x_i) = o(1)$ for all $t = O(\mathrm{poly}(k))$ and $s \neq y_i$. These two facts together allow us to control the $1 - \phi^{y_i}(g_t(x_i))$ terms that show up throughout our analysis (see Calculation A.9), while also being sufficient for showing that we get perfect training accuracy. The intuition behind these results is that, when $g_t^{y_i}(x_i) > c \log k$, we have that $\exp(g_t^{y_i}(x_i)) > k^c$ so the $1 - \phi^{y_i}(g_t(x_i))$ terms in the gradient updates quickly become small and $g_t^{y_i}$ stops growing. Throughout this part of the proof and the next,

we will use the following notation (some of which has been introduced previously) to simplify the presentation.

$$N_y = \{i : i \in [N] \text{ and } y_i = y\} \qquad P_{y,\ell}(x_i) = \{p : p \in [P] \text{ and } \langle x_i^{(p)}, v_{y,\ell} \rangle > 0\}$$

$$B_{y,\ell}^{(t)} = \{r : r \in [m] \text{ and } \langle w_{y,r}^{(t)}, v_{y,\ell} \rangle \geq \rho/\delta_1\} \tag{B.1}$$

Here, $N_y$ represents the indices corresponding to class $y$ points, $P_{y,\ell}(x_i)$ (as used in Definition 4.1) represents the patch support of the feature $v_{y,\ell}$ in $x_i$ (recall the features are orthonormal), and $B_{y,\ell}^{(t)}$ represents the set of class $y$ weights that have achieved a big enough correlation with the feature $v_{y,\ell}$ to necessarily be in the linear regime of $\widetilde{\text{ReLU}}$ on all class $y$ points at iteration $t$.

Prior to beginning our analysis of the network outputs $g_t^y$, we first prove a claim that will serve as the setting for the rest of the proof.

**Claim B.1.** With probability $1 - O(1/k)$, all of the following are (simultaneously) true for every class $y \in [k]$:

- $|N_y| = \Theta(N/k)$

- $\max_{s \in [k], r \in [m], \ell \in [2]} \langle w_{s,r}^{(0)}, v_{y,\ell} \rangle = O(\log k / \sqrt{d})$

- $\forall \ell \in [2], \max_{r \in [m]} \langle w_{y,r}^{(0)}, v_{y,\ell} \rangle = \Omega(1/\sqrt{d})$

*Proof of Claim B.1.* We prove each part of the claim in order, starting with showing that $|N_y| = \Theta(N/k)$ with the desired probability for each $y$. To see this, we note that the joint distribution of the $|N_y|$ is multinomial with uniform probability $1/k$. Now by a Chernoff bound, we have that $|N_1| = \Theta(N/k)$ with probability at least $1 - \exp(\Theta(-N/k))$. Conditioning on $|N_1| = \Theta(N/k)$, we have that the joint distribution of $|N_2|, ..., |N_k|$ is multinomial with uniform probability $1/(k-1)$, so we obtain an identical Chernoff bound for $|N_2|$. Repeating this argument and taking a union bound gives that $|N_y| = \Theta(N/k)$ for all $y \in [k]$ with probability at least $1 - k \exp(\Theta(-N/k))$.

The fact that for every $y$ we have $\max_{s \in [k], r \in [m], \ell \in [2]} \langle w_{s,r}^{(0)}, v_{y,\ell} \rangle = O(\log k / \sqrt{d})$ with probability $1 - O(1/k)$ follows from Proposition A.2. Namely, using Proposition A.2 with $t = 2\sigma \sqrt{2 \log m}$ (here $\sigma = 1/\sqrt{d}$ by our choice of initialization) yields that $\max_r \langle w_{s,r}^{(0)}, v_{y,\ell} \rangle \geq 3\sqrt{2 \log k}/\sqrt{d}$ with probability bounded above by $1/k^3$ for any $s, y$. Taking a union bound over $s, y$ then gives the result. The final fact follows by near identical logic but using Proposition A.3 (note that the correlations $\langle w_{s,r}^{(0)}, v_{y,\ell} \rangle$ are i.i.d. $\mathcal{N}(0, 1/d)$ due to the fact that the features are orthonormal and the weights themselves are i.i.d.). ∎

In everything that follows, we will **always** assume the conditions of Claim B.1 unless otherwise stated. We begin by proving a result concerning the size of softmax outputs $\phi^y(g_t(x))$ that we will repeatedly use throughout the rest of the proof.

**Claim B.2.** Consider $i \in N_y$ and suppose that both $\max_{s \in [k], r \in [m], \ell \in [2]} \langle w_{s,r}^{(t)}, v_{s,\ell} \rangle = O(\log k)$ and $\max_{s \neq y, r \in [m], \ell \in [2]} \langle w_{s,r}^{(t)}, v_{y,\ell} \rangle = O(\log(k)/\sqrt{d})$ hold true. If we have $g_t^y(x_i) \geq a \log k$ for some $a \in [0, \infty)$, then:

$$1 - \phi^y(g_t(x_i)) = \begin{cases} O\left(1/k^{a-1}\right) & \text{if } a > 1 \\ \Theta(1) & \text{otherwise} \end{cases}$$

*Proof of Claim B.2.* By assumption, all of the weight-feature correlations are $O(\log k)$ at $t$. Furthermore, for $s \neq y$, all of the off-diagonal correlations $\langle w_{s,r}^{(t)}, v_{y,\ell} \rangle$ are $O(\log(k)/\sqrt{d})$. This implies

that (using $\delta_4 = \Theta(k^{-1.5})$, $\rho = \Theta(1/k)$, $P = \Theta(k^2)$, and $\alpha = 8$):

$$g_t^s(x_i) \leq O\left(\frac{mP\delta_4^\alpha \max_{u \neq y}\left\langle w_{s,r}^{(t)}, v_{u,\ell}\right\rangle^\alpha}{\rho^{\alpha-1}}\right)$$

$$\leq O\left(\frac{k^{2+\alpha}\log(k)^\alpha}{k^{1.5\alpha}}\right) = O\left(\frac{\log(k)^\alpha}{k^3}\right)$$

$$\implies \exp(g_t^s(x_i)) \leq 1 + O\left(\frac{\log(k)^\alpha}{k^3}\right) \tag{B.2}$$

Where above we disregarded the constant number $(2C_P)$ of very low order correlations $\left\langle w_{s,r}^{(t)}, v_{y,\ell}\right\rangle$ and used the inequality that $\exp(x) \leq 1 + x + x^2$ for $x \leq 1$. Now by the assumption that $g_t^y(x_i) \geq a\log k$, we have $\exp(g_t^y(x_i)) \geq k^a$, so:

$$1 - \phi^y(g_t(x_i)) \leq 1 - \frac{k^a}{k^a + (k-1) + o(1)}$$

$$= \frac{k - 1 + o(1)}{k^a + (k-1) + o(1)} \tag{B.3}$$

From which the result follows. ∎

**Corollary B.3.** Under the same conditions as Claim B.2, for $s \neq y$, we have:

$$\phi^s(g_t(x_i)) = O\left(\frac{1}{k^{\max(a,1)}}\right)$$

*Proof of Corollary B.3.* Follows from Equations B.2 and B.3. ∎

With these softmax bounds in hand, we now show that the "diagonal" correlations $\left\langle w_{y,r}^{(t)}, v_{y,\ell}\right\rangle$ grow much more quickly than the the "off-diagonal" correlations $\left\langle w_{y,r}^{(t)}, v_{s,\ell}\right\rangle$ (where $s \neq y$). This will allow us to satisfy the conditions of Claim B.2 throughout training.

**Claim B.4.** Consider an arbitrary $y \in [k]$. Let $A \leq \rho/(\delta_2 - \delta_1)$ and let $T_A$ denote the first iteration at which $\max_{r \in [m], \ell \in [2]}\left\langle w_{y,r}^{(T_A)}, v_{y,\ell}\right\rangle \geq A$. Then we must have both that $T_A = O(\text{poly}(k))$ and that $\max_{r \in [m], s \neq y, \ell \in [2]}\left\langle w_{y,r}^{(T_A)}, v_{s,\ell}\right\rangle = O\left(\log(k)/\sqrt{d}\right)$.

*Proof of Claim B.4.* Firstly, all weight-feature correlations are $o(\rho)$ at initialization (see Claim B.1). Now for $s \neq y$ and $\left\langle w_{y,r}^{(0)}, v_{s,\ell}\right\rangle > 0$, we have for every $t$ at which $\left\langle w_{y,r}^{(t)}, v_{s,\ell}\right\rangle < \rho/(\delta_2 - \delta_1)$ that (using Calculations A.7 and A.9):

$$\left\langle -\eta \mathbf{\nabla}_{w_{y,r}^{(t)}} J(g, \mathcal{X}), v_{s,\ell}\right\rangle \leq -\frac{\eta}{N}\sum_{i \in N_s}\phi^y(g_t(x_i))\sum_{p \in P_{s,\ell}(x_i)}\frac{\beta_{i,p}^\alpha\left\langle w_{y,r}^{(t)}, v_{s,\ell}\right\rangle^{\alpha-1}}{\rho^{\alpha-1}}$$

$$+ \frac{\eta}{N}\sum_{i \in N_y}\left(1 - \phi^y(g_t(x_i))\right)\Theta\left(\frac{P\delta_4^\alpha \max_{u \neq y,\,\ell' \in [2]}\left\langle w_{y,r}^{(t)}, v_{u,\ell'}\right\rangle^{\alpha-1}}{\rho^{\alpha-1}}\right)$$

$$- \frac{\eta}{N}\sum_{i \notin N_y \cup N_s}\phi^y(g_t(x_i))\Theta\left(\frac{P\delta_3^\alpha \min_{u \neq y,\,\ell' \in [2]}\left\langle w_{y,r}^{(t)}, v_{u,\ell'}\right\rangle^{\alpha-1}}{\rho^{\alpha-1}}\right)$$

$$\leq \frac{\eta}{N}\sum_{i \in N_y}\left(1 - \phi^y(g_t(x_i))\right)\Theta\left(\frac{P\delta_4^\alpha \max_{u \neq y,\,\ell' \in [2]}\left\langle w_{y,r}^{(t)}, v_{u,\ell'}\right\rangle^{\alpha-1}}{\rho^{\alpha-1}}\right)$$

$$\tag{B.4}$$

Similarly, for $\left\langle w_{y,r}^{(0)}, v_{y,\ell} \right\rangle > 0$, we have for every $t$ at which $\left\langle w_{y,r}^{(t)}, v_{y,\ell} \right\rangle < \rho/(\delta_2 - \delta_1)$ that:

$$
\left\langle -\eta \boldsymbol{\nabla}_{w_{y,r}^{(t)}} J(g, \mathcal{X}), v_{y,\ell} \right\rangle \geq \frac{\eta}{N} \sum_{i \in N_y} \left(1 - \phi^y\big(g_t(x_i)\big)\right) \sum_{p \in P_{s,\ell}(x_i)} \frac{\beta_{i,p}^\alpha \left\langle w_{y,r}^{(t)}, v_{y,\ell} \right\rangle^{\alpha-1}}{\rho^{\alpha-1}}
$$
$$
- \frac{\eta}{N} \sum_{i \notin N_y} \phi^y\big(g_t(x_i)\big) \Theta \left( \frac{P\delta_4^\alpha \max_{u \neq y,\ \ell' \in [2]} \left\langle w_{y,r}^{(t)}, v_{u,\ell'} \right\rangle^{\alpha-1}}{\rho^{\alpha-1}} \right)
$$
(B.5)

From Equation B.5, Claim B.2, and Corollary B.3 we get that for $t \leq T_A$:

$$
\left\langle -\eta \boldsymbol{\nabla}_{w_{y,r}^{(t)}} J(g, \mathcal{X}), v_{y,\ell} \right\rangle \geq \Theta \left( \frac{\eta \left\langle w_{y,r}^{(t)}, v_{y,\ell} \right\rangle^{\alpha-1}}{k \rho^{\alpha-1}} \right)
$$
(B.6)

Where above we also used the fact that $|N_y| = \Theta(N/k)$. On the other hand, also using Claim B.2 and Corollary B.3, we have that for all $t$ for which $\left\langle w_{y,r}^{(t)}, v_{s,\ell} \right\rangle < \rho/(\delta_2 - \delta_1)$:

$$
\left\langle -\eta \boldsymbol{\nabla}_{w_{y,r}^{(t)}} J(g, \mathcal{X}), v_{s,\ell} \right\rangle \leq \Theta \left( \frac{\eta P \delta_4^\alpha \max_{u \neq y,\ \ell' \in [2]} \left\langle w_{y,r}^{(t)}, v_{u,\ell'} \right\rangle^{\alpha-1}}{\rho^{\alpha-1}} \right)
$$
(B.7)

Now suppose that $\left\langle w_{y,r}^{(0)}, v_{s,\ell} \right\rangle$ is the maximum off-diagonal correlation at initialization. Then using Equation B.7, we can lower bound the number of iterations $T$ it takes for $\left\langle w_{y,r}^{(t)}, v_{s,\ell} \right\rangle$ to grow by a fixed constant $C$ factor from initialization:

$$
T\Theta \left( \frac{\eta P \delta_4^\alpha C^{\alpha-1} \left\langle w_{y,r}^{(0)}, v_{s,\ell} \right\rangle^{\alpha-1}}{\rho^{\alpha-1}} \right) \geq (C-1) \left\langle w_{y,r}^{(0)}, v_{s,\ell} \right\rangle
$$
$$
\implies T \geq \Theta \left( \frac{\rho^{\alpha-1}}{\eta P \delta_4^\alpha \left\langle w_{y,r}^{(0)}, v_{s,\ell} \right\rangle^{\alpha-2}} \right)
$$
$$
= \Theta \left( \frac{k^{1.5\alpha-2} \rho^{\alpha-1} d^{\alpha/2-1}}{\eta} \right)
$$
(B.8)

As there exists at least one $\left\langle w_{y,r}^{(0)}, v_{y,\ell} \right\rangle = \Omega(1/\sqrt{d})$, it immediately follows from comparing to Equation B.6 and recalling that $\alpha = 8$ in Assumption 4.3 that $T >> T_A$, and that $T_A = O(\text{poly}(k))$, so the claim is proved. $\blacksquare$

Having established strong control over the off-diagonal correlations, we are now ready to prove the first half of the main result of this part of the proof - that $g_t^y(x_i)$ reaches $\Omega(\log k)$ for all $i \in N_y$ in $O(\text{poly}(k))$ iterations. In proving this, it will help us to have some control over the network outputs $g_t^y$ across different points $x_i$ and $x_j$ at the later stages of training, which we take care of below.

**Claim B.5.** For every $y \in [k]$ and all $t$ such that $\max_{i \in N_y} g_t^y(x_i) \geq \log k$ and $\max_{r \in [m], s \neq y, \ell \in [2]} \left\langle w_{y,r}^{(t)}, v_{s,\ell} \right\rangle = O\left(\log(k)/\sqrt{d}\right)$, we have $\max_{i \in N_y} g_t^y(x_i) = \Omega(\min_{i \in N_y} g_t^y(x_i))$.

*Proof of Claim B.5.* Let $j = \text{argmax}_{i \in N_y} g_t^y(x_i)$. Since $g_t^y(x_j) \geq \log k$, we necessarily have that $B_{y,\ell}$ is non-empty for at least one $\ell \in [2]$ (since $m\rho = \Theta(1)$). Only those weights $w_{y,r}^{(t)}$ with $r \in B_{y,\ell}$

for some $\ell \in [2]$ are asymptotically relevant (as any weights not considered can only contribute a $O(1)$ term), and we can write:

$$g_t^y(x_j) \leq \sum_{\ell \in [2]} \left( \sum_{p \in P_{y,\ell}(x_i)} \beta_{i,p} \right) \sum_{r \in B_{y,\ell}^{(t)}} \left\langle w_{y,r}^{(t)}, v_{y,\ell} \right\rangle + o(\log k)$$

For any other $j \in N_y$, we have that $\beta_{j,p} \geq \delta_1 \beta_{i,p}/(\delta_2 - \delta_1)$, from which the result follows. ∎

Now we may show:

**Claim B.6.** For each $y \in [k]$, let $T_y$ denote the first iteration such that $\max_{i \in N_y} g_{T_y}^y(x_i) \geq \log k$. Then $T_y = O(\text{poly}(k))$ and $\max_{r \in [m], s \neq y, \ell \in [2]} \left\langle w_{y,r}^{(T_y)}, v_{s,\ell} \right\rangle = O\left(\log(k)/\sqrt{d}\right)$. Furthermore, $\min_{i \in N_y} g_t^y(x_i) = \Omega(\log k)$ for all $t \geq T_y$.

*Proof of Claim B.6.* Applying Claim B.4 to an arbitrary $y \in [k]$ yields the existence of a correlation $\left\langle w_{y,r^*}^{(t)}, v_{y,\ell^*} \right\rangle \geq \rho/(\delta_2 - \delta_1)$. Reusing the logic of Claim B.4, but this time replacing $\left\langle w_{y,r^*}^{(t)}, v_{y,\ell^*} \right\rangle$ in Equation B.6 with $\rho/(\delta_2 - \delta_1)$, shows that in $O(\text{poly}(k))$ additional iterations we have $\left\langle w_{y,r^*}^{(t)}, v_{y,\ell^*} \right\rangle \geq \rho/\delta_1$ (implying $r \in B_{y,\ell^*}^{(t)}$) while the off-diagonal correlations still remain within a constant factor of initialization.

Now we may lower bound the update to $\left\langle w_{y,r^*}^{(t)}, v_{y,\ell^*} \right\rangle$ as (using Calculation A.8):

$$\left\langle -\eta \boldsymbol{\nabla}_{w_{y,r^*}^{(t)}} J(g, \mathcal{X}), v_{y,\ell^*} \right\rangle \geq \frac{\eta}{N} \sum_{i \in N_y} \left( 1 - \phi^y\big(g_t(x_i)\big) \right) \sum_{p \in P_{s,\ell}(x_i)} \beta_{i,p}$$
$$- \frac{\eta}{N} \sum_{i \notin N_y} \phi^y\big(g_t(x_i)\big) \Theta \left( \frac{P \delta_4^\alpha \max_{u \neq y,\, \ell' \in [2]} \left\langle w_{y,r^*}^{(t)}, v_{u,\ell'} \right\rangle^{\alpha-1}}{\rho^{\alpha-1}} \right) \tag{B.9}$$

So long as $\max_{i \in N_y} g_t^y(x_i) < \log k$ (which is necessarily still the case at this point, as again $m\rho = \Theta(1)$), we have by the logic of Claim B.2 that we can simplify Equation B.9 to:

$$\left\langle -\eta \boldsymbol{\nabla}_{w_{y,r^*}^{(t)}} J(g, \mathcal{X}), v_{y,\ell^*} \right\rangle \geq \Theta(\eta/k) \tag{B.10}$$

Where again we used the fact that $|N_y| = \Theta(N/k)$. Now we can upper bound $T_y$ by the number of iterations it takes $\left\langle w_{y,r^*}^{(t)}, v_{y,\ell^*} \right\rangle$ to grow to $\log(k)/\delta_1$. From Equation B.10, we clearly have that $T_y = O(\text{poly}(k))$ for some polynomial in $k$. Furthermore, comparing to Equation B.7, we necessarily still have $\max_{r \in [m], s \neq y, \ell \in [2]} \left\langle w_{y,r}^{(T_y)}, v_{s,\ell} \right\rangle = O\left(\log(k)/\sqrt{d}\right)$. Finally, as the update in Equation B.9 is positive at $T_y$ (and the absolute value of a gradient update is $o(1)$), it follows that $\min_{i \in N_y} g_t^y(x_i) = \Omega(\log k)$ for all $t \geq T_y$ by Claim B.5. ∎

The final remaining task is to show that $g_t^y(x_i) = O(\log k)$ and $g_t^s(x_i) = o(1)$ for all $t = O(\text{poly}(k))$ and $i \in N_y$ for every $y \in [k]$.

**Claim B.7.** For all $t = O(k^C)$ for any universal constant $C$, and for every $y \in [k]$ and $s \neq y$, we have that $g_t^y(x_i) = O(\log k)$ and $\max_{r \in [m], s \neq y, \ell \in [2]} \left\langle w_{y,r}^{(t)}, v_{s,\ell} \right\rangle = O\left(\log(k)/\sqrt{d}\right)$ for all $i \in N_y$.

*Proof of Claim B.7.* Let us again consider any class $y \in [k]$ and $t \geq T_y$. The idea is to show that $\max_{i \in N_y} 1 - \phi^y(g_t(x_i))$ is decreasing rapidly as $\min_{i \in N_y} g_t^y(x_i)$ grows to successive levels of $a \log k$ for $a > 1$.

Firstly, following Equation B.9, we can form the following upper bound for the gradient updates to $r \in B_{y,\ell}^{(t)}$:

$$\left\langle -\eta \boldsymbol{\nabla}_{w_{y,r^*}^{(t)}} J(g, \mathcal{X}), v_{y,\ell^*} \right\rangle \le \frac{\eta}{N} \sum_{i \in N_y} \left( 1 - \phi^y \big( g_t(x_i) \big) \right) \sum_{p \in P_{s,\ell}(x_i)} \beta_{i,p}$$

$$\le \left( 1 - \min_{i \in N_y} \phi^y \big( g_t(x_i) \big) \right) \Theta(\eta/k) \qquad \text{(B.11)}$$

From Equation B.11 it follows that it takes at least $\Theta(k \log(k)/(m\eta))$ iterations (since the correlations must grow at least $\log(k)/m$ from $T_y$ for $g_t^y(x_i)$ to reach $2 \log k$. Now let $T_a$ denote the number of iterations it takes for $\min_{i \in N_y} g_t^y(x_i)$ to cross $a \log k$ *after* crossing $(a-1) \log k$ for the first time. For $a \ge 3$, we necessarily have that $T_a = \Omega(k T_{a-1})$ by Claim B.2 and Equation B.11.

Let us now further define $T_f$ to be the first iteration at which $\max_{i \in N_y} g_{T_f}^y(x_i) \ge f(k) \log k$ for some $f(k) = \omega(1)$. By Claim B.5, at this point $\min_{i \in N_y} g_{T_f}^y(x_i) = \Omega(f(k) \log k)$. However, we have from the above discussion that:

$$T_f \ge \Omega(\text{poly}(k)) + \sum_{a=0}^{f(k)-3} \Omega\left( \frac{k^a \log k}{\eta} \right)$$

$$\ge \Omega\left( \frac{\log k \left( k^{f(k)-2} - 1 \right)}{\eta(k-1)} \right)$$

$$\ge \omega(\text{poly}(k)) \qquad \text{(B.12)}$$

So $\max_{i \in N_y} g_t^y(x_i) = O(\log k)$ for all $t = O(\text{poly}(k))$. An identical analysis also works for the off-diagonal correlations $\left\langle w_{y,r}^{(t)}, v_{s,\ell} \right\rangle$ but forming an upper bound using Equation B.4, so we are done. ∎

We get the following two corollaries as straightforward consequences of Claim B.7.

**Corollary B.8** (Perfect Training Accuracy). We have that there exists a universal constant $C$ such that $\text{argmax}_s g_t^s(x_i) = y_i$ for every $(x_i, y_i) \in \mathcal{X}$ for all $t \ge k^C$ but with $t = O(\text{poly}(k))$.

**Corollary B.9** (Softmax Control). We have that for all $y \in [k]$ and any $t = O(\text{poly}(k))$ for any polynomial in $k$ that $\max_{i \in N_y} \sum_{s \ne y} \exp(g_t^s(x_i)) = k + o(1)$.

Corollary B.8 finishes this part of the proof.

**Part II.**

For the next part of the proof, we characterize the separation between $\max_{r \in [m]} \left\langle w_{y,r}^{(0)}, v_{y,1} \right\rangle$ and $\max_{r \in [m]} \left\langle w_{y,r}^{(0)}, v_{y,2} \right\rangle$, and show that this separation (when it is significant enough) gets amplified over the course of training. To show this, we will rely largely on the techniques found in Allen-Zhu and Li (2021), and finish in a near-identical manner to the proof of Claim B.7.

As with Part I, we first introduce some notation that we will use throughout this part of the proof.

$$S_{y,\ell} = \frac{1}{N} \sum_{i \in N_y} \sum_{p \in P_{y,\ell}(x_i)} \beta_{i,p}^\alpha \qquad \Lambda_{y,\ell}^{(t)} = \max_{r \in [m]} \left\langle w_{y,r}^{(t)}, v_{y,\ell} \right\rangle$$

Here, $S_{y,\ell}$ represents the data-dependent quantities that show up in the gradient updates to the correlations during the phase of training in which the correlations are in the polynomial part of $\widetilde{\text{ReLU}}$, while $\Lambda_{y,\ell}^{(t)}$ represents the max class $y$ correlation with feature $v_{y,\ell}$ at time $t$.

Now we can prove essentially the same result as Proposition B.2 in Allen-Zhu and Li (2021), which quantifies the separation between $\Lambda_{y,1}^{(0)}$ and $\Lambda_{y,2}^{(0)}$ after taking into account $S_{y,1}$ and $S_{y,2}$.

**Claim B.10** (Feature Separation at Initialization). For each class $y$, we have that either:

$$\Lambda_{y,1}^{(0)} \geq \left(\frac{S_{y,2}}{S_{y,1}}\right)^{\frac{1}{\alpha-2}} \left(1 + \Theta\left(\frac{1}{\log^2 k}\right)\right) \Lambda_{y,2}^{(0)} \quad \text{or}$$

$$\Lambda_{y,2}^{(0)} \geq \left(\frac{S_{y,1}}{S_{y,2}}\right)^{\frac{1}{\alpha-2}} \left(1 + \Theta\left(\frac{1}{\log^2 k}\right)\right) \Lambda_{y,1}^{(0)}$$

with probability $1 - O\left(\frac{1}{\log k}\right)$.

*Proof of Claim B.10.* Suppose WLOG that $S_{y,1} \geq S_{y,2}$. If neither of the inequalities in the claim hold, then we have that:

$$\Lambda_{y,1}^{(0)} \in \left(\frac{S_{y,2}}{S_{y,1}}\right)^{\frac{1}{\alpha-2}} \left(1 \pm \Theta\left(\frac{1}{\log^2 k}\right)\right) \Lambda_{y,2}^{(0)}$$

Which follows from the fact that, for a constant $A$, we have:

$$\frac{1}{1 + \frac{A}{\log^2 k}} \geq 1 - \frac{A}{\log^2 k}$$

Now we recall that $\Lambda_{y,1}^{(0)}$ and $\Lambda_{y,2}^{(0)}$ are both maximums over i.i.d. $\mathcal{N}(0, 1/d)$ variables (again, since the feature vectors are orthonormal), so we can apply Corollary A.5 (Gaussian anti-concentration) to $\Lambda_{y,1}^{(0)}$ while taking $\epsilon = (S_{y,2}/S_{y,1})^{\frac{1}{\alpha-2}} \Theta\left(1/\log^2 k\right) \Lambda_{y,2}^{(0)}$ and $x = (S_{y,2}/S_{y,1})^{\frac{1}{\alpha-2}} \Lambda_{y,2}^{(0)}$. It is crucial to note that we can only do this because $\Lambda_{y,2}^{(0)}$ is independent of $\Lambda_{y,1}^{(0)}$, and both take values over all of $\mathbb{R}$. From this we get that:

$$P\left(\Lambda_{y,1}^{(0)} \in (S_{y,2}/S_{y,1})^{\frac{1}{\alpha-2}} \Lambda_{y,2}^{(0)} \pm \epsilon\right) \leq \frac{4\epsilon(1 + \sqrt{2\log m})}{\sigma}$$

$$= O\left(\frac{\sigma\sqrt{\log m}}{\log^2 k}\right) \Theta\left(\frac{\sqrt{\log m}}{\sigma}\right)$$

$$= O\left(\frac{1}{\log k}\right) \quad \text{with probability } 1 - \frac{1}{m}$$

Where we used the fact that $m = \Theta(k)$ and Proposition A.2 to characterize $\Lambda_{y,2}^{(0)}$ (also noting that $S_{y,2}/S_{y,1}$ is $\Theta(1)$). Thus, neither of the inequalities hold with probability $O(1/\log k)$, so we have the desired result. ∎

We can use the separation from Claim B.10 to show that, in the initial stages of training, the max correlated weight/feature pair grows out of the polynomial region of $\widetilde{\text{ReLU}}$ and becomes large much faster than the correlations with the other feature for the same class. For $y \in [k]$, let $\ell^*$ be such that $\Lambda_{y,\ell^*}^{(0)}$ is the left-hand side of the satisfied inequality from Claim B.10. Additionally, let $r^* = \text{argmax}_r \left\langle w_{y,r}^{(0)}, v_{y,\ell^*}\right\rangle$, i.e. the strongest weight/feature correlation pair at initialization. We will show that when $\left\langle w_{y,r^*}^{(t)}, v_{y,\ell^*}\right\rangle$ becomes $\Omega(\rho)$, the other correlations remain small. In order to do so, we need a useful lemma from Allen-Zhu and Li (2021) that we restate below.

**Lemma B.11** (Lemma C.19 from Allen-Zhu and Li (2021)). Let $q \geq 3$ be a constant and $x_0, y_0 = o(1)$. Let $\{x_t, y_t\}_{t \geq 0}$ be two positive sequences updated as

- $x_{t+1} \geq x_t + \eta C_t x_t^{q-1}$ for some $C_t = \Theta(1)$, and

- $y_{t+1} \leq y_t + \eta S C_t y_t^{q-1}$ for some constant $S = \Theta(1)$.

Where $\eta = O(1/\text{poly}(k))$ for a sufficiently large polynomial in $k$. Suppose $x_0 \geq y_0 S^{\frac{1}{q-2}}\left(1 + \Theta\left(\frac{1}{\text{polylog}(k)}\right)\right)$. For every $A = O(1)$, letting $T_x$ be the first iteration such that $x_t \geq A$, we must have that

$$y_{T_x} = O(y_0 \text{polylog}(k))$$

To apply Lemma B.11 in our setting, we first prove the following claim.

**Claim B.12.** For a class $y \in [k]$, we define the following two sequences:

$$a_{y,t} = \left(\frac{S_{y,\ell^*}}{\rho^{\alpha-1}}\right)^{\frac{1}{\alpha-2}} \left\langle w_{y,r^*}^{(t)}, v_{y,\ell^*} \right\rangle \quad \text{and} \quad b_{y,t} = \left(\frac{S_{y,3-\ell^*}}{\rho^{\alpha-1}}\right)^{\frac{1}{\alpha-2}} \left\langle w_{y,r}^{(t)}, v_{y,3-\ell^*} \right\rangle$$

Where the $r$ in the definition of $b_{y,t}$ is arbitrary. Then with probability $1 - O\left(\frac{1}{\log k}\right)$ there exist $C_t, S = \Theta(1)$ such that for all $t$ for which $\left\langle w_{y,r^*}^{(t)}, v_{y,\ell^*} \right\rangle < \rho/(\delta_2 - \delta_1)$:

$$a_{y,t+1} \geq a_{y,t} + \eta C_t a_{y,t}^{\alpha-1}$$
$$b_{y,t+1} \leq b_{y,t} + \eta S C_t b_{y,t}^{\alpha-1}$$

Additionally (with the same probability), we have that $a_{y,0} \geq S^{\frac{1}{\alpha-2}}\left(1 + \Theta\left(\frac{1}{\text{polylog}(k)}\right)\right) b_{y,0}$.

*Proof of Claim B.12.* The update to $\left\langle w_{y,r^*}^{(t)}, v_{y,\ell^*} \right\rangle$ in this regime can be bounded as follows (using Corollary B.9 and recalling Equation B.5):

$$\left\langle -\eta \nabla_{w_{y,r^*}^{(t)}} J(g,\mathcal{X}), v_{y,\ell^*} \right\rangle \geq \frac{\eta}{N} \sum_{i \in N_y} \left(1 - \phi^y(g_t(x_i))\right) \sum_{p \in P_{s,\ell}(x_i)} \frac{\beta_{i,p}^{\alpha} \left\langle w_{y,r}^{(t)}, v_{y,\ell} \right\rangle^{\alpha-1}}{\rho^{\alpha-1}}$$

$$- \frac{\eta}{N} \sum_{i \notin N_y} \phi^y(g_t(x_i)) \Theta\left(\frac{P\delta_4^{\alpha} \max_{u \neq y, \, \ell' \in [2]} \left\langle w_{y,r}^{(t)}, v_{u,\ell'} \right\rangle^{\alpha-1}}{\rho^{\alpha-1}}\right)$$

$$\geq \eta\left(1 - \Theta\left(\frac{1}{k}\right)\right) S_{y,\ell^*} \frac{\left\langle w_{y,r^*}^{(t)}, v_{y,\ell^*} \right\rangle^{\alpha-1}}{\rho^{\alpha-1}} \tag{B.13}$$

Similarly, we have (noting also that $\left\langle w_{y,r}^{(t)}, v_{y,3-\ell^*} \right\rangle < \rho/(\delta_2 - \delta_1)$):

$$\left\langle -\eta \nabla_{w_{y,r}^{(t)}} J(g,\mathcal{X}), v_{y,3-\ell^*} \right\rangle \leq \eta S_{y,3-\ell^*} \frac{\left\langle w_{y,r}^{(t)}, v_{y,3-\ell^*} \right\rangle^{\alpha-1}}{\rho^{\alpha-1}} \tag{B.14}$$

Multiplying the above inequalities by $\left(S_{y,\ell^*}/\rho^{\alpha-1}\right)^{\frac{1}{\alpha-2}}$, we see that $a_{y,t}$ and $b_{y,t}$ satisfy the inequalities in the claim with $C_t = 1 - \Theta\left(\frac{1}{k}\right)$ and $S = \left(S_{y,3-\ell^*}/S_{y,\ell^*}\right)\left(1 + \Theta\left(\frac{1}{k}\right)\right)$. Now by Claim B.10 we have:

$$a_{y,0} \geq \left(\frac{S_{y,3-\ell^*}}{S_{y,\ell^*}}\right)^{\frac{1}{\alpha-2}}\left(1 + \Theta\left(\frac{1}{\log^2 k}\right)\right) b_{y,0}$$

$$\geq S^{\frac{1}{\alpha-2}}\left(1 + \Theta\left(\frac{1}{\text{polylog}(k)}\right)\right) b_{y,0}$$

So we are done. ∎

Now by the fact that $|N_y| = \Theta(N/k)$, we have $S_{y,1}, S_{y,2} = \Theta(1/k) = O(\rho)$, which implies that $\left(S_{y,\ell^*}/\rho^{\alpha-1}\right)^{1/(\alpha-2)} = O(1/\rho)$. From this we get that while $a_{y,t} < C/(\delta_2 - \delta_1)$ for some appropriately chosen constant $C$, we have $\left\langle w_{y,r^*}^{(t)}, v_{y,\ell^*} \right\rangle < \rho/(\delta_2 - \delta_1)$.

Since Claim B.12 holds in this regime, we can apply Lemma B.11 with $A = C/(\delta_2 - \delta_1)$, which gives us that when $a_{y,t} \geq C/(\delta_2 - \delta_1)$, we have $b_{y,t} = O(b_{y,0}\text{polylog}(k))$. From this we obtain that when $\left\langle w_{y,r^*}^{(t)}, v_{y,\ell^*} \right\rangle \geq \rho/(\delta_2 - \delta_1)$ we have that $\left\langle w_{y,r}^{(t)}, v_{y,3-\ell^*} \right\rangle$ is still within a $\text{polylog}(k)$ factor of $\left\langle w_{y,r}^{(0)}, v_{y,3-\ell^*} \right\rangle$ for any $r$.

Now from the same logic as the proof of Claim B.7, we can show that this separation remains throughout training.

**Claim B.13.** For any class $y \in [k]$, with probability $1 - O(1/\log k)$, we have that $\max_{r \in [m]} \left\langle w_{y,r}^{(t)}, v_{y,3-\ell^*} \right\rangle = O(\text{polylog}(k) \max_{r \in [m]} \left\langle w_{y,r}^{(0)}, v_{y,3-\ell^*} \right\rangle)$ for all $t = O(\text{poly}(k))$ for any polynomial in $k$.

*Proof of Claim B.13.* It follows from the same logic as in the proof of Claim B.6 that at the first iteration $t$ for which we have $\min_{i \in N_y} g_t^y(x_i) \geq \log k$, we still have $\left\langle w_{y,r}^{(t)}, v_{y,3-\ell^*} \right\rangle$ is within some polylog$(k)$ factor of initialization (here the correlation $\left\langle w_{y,r}^{(t)}, v_{y,3-\ell^*} \right\rangle$ can be viewed as the same as an off-diagonal correlation from the proof of Claim B.6). The rest of the proof then follows from identical logic to that of Claim B.7; namely, we can show that for $\left\langle w_{y,r}^{(t)}, v_{y,3-\ell^*} \right\rangle$ to grow by more than a polylog$(k)$ factor we need $\omega(\text{poly}(k))$ training iterations. ∎

From Claim B.13 along with Claim B.7, it follows that with probability $1 - O(1/\log k)$, for any class $y$ (after polynomially many training iterations) we have:

$$g_t^y(x_i') = O\left( \frac{m\text{polylog}(k)}{\sqrt{d}} \right) = O\left( \frac{k\text{polylog}(k)}{\sqrt{d}} \right)$$

$$g_t^s(x_i') = \Omega\left( P \frac{\log(k)^\alpha}{\rho^{\alpha-1}k^{2.5\alpha}} \right) = \omega\left( \frac{k\text{polylog}(k)}{\sqrt{d}} \right) \quad \text{for } s \neq y, \text{ if } \exists P_{s,\ell}(x_i) \neq \emptyset \qquad (B.15)$$

Where $x_i'$ is any point $x_i$ with $i \in N_y$ modified so that all instances of feature $v_{y,\ell^*}$ are replaced by 0, and the second line above follows from the fact that by Claim B.7 we must have $\left\langle w_{s,r}^{(t)}, v_{s,\ell} \right\rangle = \Omega(\log(k)/k)$ for at least some $r, \ell$ for every $s$ (and $d = \Theta(k^{20})$). This proves that feature $v_{y,3-\ell^*}$ is not learned in the sense of Definition 4.5.

Using Claim B.13 for each class, we have by a Chernoff bound that with probability at least $1 - o(1/k)$ that for $(1 - o(1))k$ classes only a single feature is learned, which proves the theorem. □

## B.2 Proof of Theorem 4.7

**Theorem 4.7.** For $k$ and $N$ sufficiently large and the settings stated in Assumptions 4.3, we have that the following hold with probability $1 - O(1/k)$ after running gradient descent with a step size $\eta = O(1/\text{poly}(k))$ for $O(\text{poly}(k)/\eta)$ iterations on $J_{MM}(g, \mathcal{X})$ (for sufficiently large polynomials in $k$):

1. (Training accuracy is perfect): For all $(x_i, y_i) \in \mathcal{X}$, we have $\text{argmax}_s g^s(x_i) = y_i$.

2. (Both features are learned): For each class $y \in [k]$, both $v_{y,1}$ and $v_{y,2}$ are learned in the sense of Definition 4.5 by the model $g$.

Furthermore, the above remains true for all $t = O(\text{poly}(k))$ for any polynomial in $k$.

*Proof.* As in the proof of Theorem 4.6, we break the proof into two parts. The first part mirrors most of the structure of Part I of the proof of Theorem 4.6, in that we analyze the off-diagonal correlations and also show that the network outputs $g_t^y$ can grow to (and remain) $\Omega(\log k)$ as training progresses. However, we do not show that the outputs stay $O(\log k)$ in Part I (as we did in the ERM case), as there are additional subtleties in the Midpoint Mixup analysis that require different techniques which we find are more easily introduced separately.

The second part of the proof differs significantly from Part II of the proof of Theorem 4.6, as our goal is to now show that any large separation between weight-feature correlations for each class are corrected over the course of training. At a high level, we show this by proving a gradient correlation lower bound that depends only on the magnitude of the separation between correlations and the variance of the feature coefficients in the data distribution, after which we can conclude that any feature lagging behind will catch up in polynomially many training iterations. We then use the techniques from the gradient lower bound analysis to prove that the network outputs $g_t^y$ stay $O(\log k)$ throughout training, which wraps up the proof.

**Part I.** We first recall that $z_{i,j} = (x_i + x_j)/2$, and we refer to such $z_{i,j}$ as "mixed points". In this part of the proof, we show that $g_t^{y_i}(z_{i,j})$ crosses $\log k$ on at least one mixed point $z_{i,j}$ in polynomially many iterations (after which the network outputs remain $\Omega(\log k)$). As before, this requires getting a handle on the off-diagonal correlations $\left\langle w_{y,r}^{(t)}, v_{s,\ell} \right\rangle$ (with $s \neq y$).

Throughout the proof, we will continue to rely on the notation introduced in Equation B.1 in the proof of Theorem 4.6. However, we make one slight modification to the definition of $B_{y,\ell}^{(t)}$ for the Mixup case (so as to be able to handle mixed points), which is as follows:

$$B_{y,\ell}^{(t)} = \{r : r \in [m] \text{ and } \left\langle w_{y,r}^{(t)}, v_{y,\ell} \right\rangle \geq 2\rho/\delta_1\} \tag{B.16}$$

We again start by proving a claim that will constitute our setting for the rest of this proof.

**Claim B.14.** With probability $1 - O(1/k)$, all of the following are (simultaneously) true for every class $y \in [k]$:

- $|N_y| = \Theta(N/k)$

- $\max_{s \in [k], r \in [m], \ell \in [2]} \left\langle w_{s,r}^{(0)}, v_{y,\ell} \right\rangle = O(\log k/\sqrt{d})$

- $\forall \ell \in [2], \ \max_{r \in [m]} \left\langle w_{y,r}^{(0)}, v_{y,\ell} \right\rangle = \Omega(1/\sqrt{d})$

- For $\Omega(k)$ tuples $(s, \ell) \in [k] \times [2]$ we have $\left\langle w_{y,r}^{(0)}, v_{s,\ell} \right\rangle > 0$.

*Proof of Claim B.14.* The first three items in the claim are exactly the same as in Claim B.1, and the last item is true because the correlations $\left\langle w_{y,r}^{(0)}, v_{s,\ell} \right\rangle$ are mean zero Gaussians. ∎

Once again, in everything that follows, we will **always** assume the conditions of Claim B.14 unless otherwise stated. We now translate Claim B.2 to the Midpoint Mixup setting.

**Claim B.15.** Consider $i \in N_y$, $j \in N_s$ for $s \neq y$ and suppose that $\max_{u \notin \{y,s\}} g_t^u(z_{i,j}) = O(\log(k)/k)$ holds true. If we have $g_t^y(z_{i,j}) = a \log k$ and $g_t^s(z_{i,j}) = b \log k$ for $a, b = O(1)$, then:

$$1 - 2\phi^y(g_t(z_{i,j})) = \begin{cases} -\Omega(1) & \text{if } a > 1, \ a - b = \Omega(1) \\ \pm O(1) & \text{if } a > 1, \ a - b = \pm o(1) \\ \Theta(1) & \text{otherwise} \end{cases}$$

Where in the second item above the sign of $1 - 2\phi^y(g_t(z_{i,j}))$ depends on the sign of $a - b$.

*Proof of Claim B.15.* In comparison to Claim B.2, the Midpoint Mixup case is slightly more involved in that $g_t^s(z_{i,j})$ can be quite large due to the $x_j$ part of $z_{i,j}$. As a result, we directly assume some control over the different class outputs on the mixed points (which we will prove to hold throughout training later). By assumption, we have for $u \neq y, s$:

$$g_t^u(z_{i,j}) = O(\log(k)/k) \implies \exp(g_t^u(z_{i,j})) \leq 1 + O(\log(k)/k) \tag{B.17}$$

Where above we used the inequality $\exp(x) \leq 1 + x + x^2$ for $x \in [0, 1]$. Now by the assumptions that $g_t^y(z_{i,j}) = a \log k$ and $g_t^s(z_{i,j}) = b \log k$, we have:

$$\begin{aligned} 1 - 2\phi^y(g_t(x_i)) &\leq 1 - \frac{2k^a}{k^a + k^b + (k-2) + o(k)} \\ &= \frac{k^b - k^a + (k-2) + o(k)}{k^a + k^b + (k-2) + o(k)} \end{aligned} \tag{B.18}$$

From which the result follows. ∎

**Corollary B.16.** Under the same conditions as Claim B.15, for $u \neq y, s$ we have:

$$1 - 2\phi^u(g_t(z_{i,j})) = \Theta(1)$$

*Proof of Corollary B.16.* Follows from Equations B.17 and B.18. ∎

We observe that Claim B.15 and Corollary B.16 are less precise than Claim B.2, largely because there is now a dependence on the gap between the class $y$ and class $s$ network outputs as opposed to just the class $y$ network output. We are now again ready to compare the growth of the diagonal correlations $\left\langle w_{y,r}^{(t)}, v_{y,\ell} \right\rangle$ with the off-diagonal correlations $\left\langle w_{y,r}^{(t)}, v_{s,\ell} \right\rangle$. However, this is not as straightforward as it was in the ERM setting. The issue is that the off-diagonal correlations can actually grow significantly, due to the fact that the features $v_{y,\ell}$ can show up when mixing points in class $y$ with class $s$.

**Claim B.17.** Fix an arbitrary class $y \in [k]$. Let $A \in [\Omega(\rho), \rho/(\delta_2 - \delta_1)]$ and let $T_A$ be the first iteration at which $\max_{r \in [m], \ell \in [2]} \left\langle w_{y,r}^{(T_A)}, v_{y,\ell} \right\rangle \geq A$; we must have both that $T_A = O(\text{poly}(k))$ and that, for every $s \neq y$ and $\ell \in [2]$:

$$\left\langle w_{y,r}^{(T_A)}, v_{s,\ell} \right\rangle = O\left( \max_{\ell' \in [2]} \left\langle w_{y,r}^{(T_A)}, v_{y,\ell'} \right\rangle / k \right)$$

Additionally, for all $s, \ell$ with $\left\langle w_{y,r}^{(0)}, v_{s,\ell} \right\rangle > 0$, we have that $\left\langle w_{y,r}^{(T_A)}, v_{s,\ell} \right\rangle = \Omega\left( \frac{\left\langle w_{y,r}^{(0)}, v_{s,\ell} \right\rangle}{\text{polylog}(k)} \right)$.

*Proof of Claim B.17.* By our setting, we must have that there exists a diagonal correlation $\left\langle w_{y,r^*}^{(0)}, v_{y,\ell^*} \right\rangle = \Omega(1/\sqrt{d})$, which we will focus our attention on. Using Calculation A.10 and the ideas from Calculation A.7, we can lower bound the update to $\left\langle w_{y,r^*}^{(t)}, v_{y,\ell^*} \right\rangle$ from initialization up to $T_A$ as:

$$\begin{aligned}
\left\langle -\eta \boldsymbol{\nabla}_{w_{y,r^*}^{(t)}} J_{MM}(g, \mathcal{X}), v_{y,\ell^*} \right\rangle &\geq \frac{\eta}{N^2} \sum_{i \in N_y} \sum_{j \notin N_y} \left( 1 - 2\phi^y\big((g_t(z_{i,j}))\big) \right) \Theta\left( \frac{\left\langle w_{y,r^*}^{(t)}, v_{y,\ell^*} \right\rangle^{\alpha-1}}{\rho^{\alpha-1}} \right) \\
&\quad + \frac{\eta}{N^2} \sum_{i \in N_y} \sum_{j \in N_y} \left( 1 - \phi^y\big((g_t(z_{i,j}))\big) \right) \Theta\left( \frac{\left\langle w_{y,r^*}^{(t)}, v_{y,\ell^*} \right\rangle^{\alpha-1}}{\rho^{\alpha-1}} \right) \\
&\quad - \frac{\eta}{N^2} \sum_{i \notin N_y} \sum_{j \notin N_y} \phi^y\big(g_t(z_{i,j})\big) \Theta\left( \frac{P\sigma_4^\alpha \max_{u \in [k], q \in [2]} \left\langle w_{y,r^*}^{(t)}, v_{u,q} \right\rangle^{\alpha-1}}{\rho^{\alpha-1}} \right)
\end{aligned}$$

$$\tag{B.19}$$

Above we made use of the fact that, for $i \in N_y$ and $j \notin N_y$, we have $\left\langle w_{y,r^*}^{(t)}, z_{i,j}^{(p)} \right\rangle \geq \left\langle w_{y,r^*}^{(t)}, v_{y,\ell^*} \right\rangle / 2$ for at least $\Theta(|N_y|N)$ mixed points since the correlation $\left\langle w_{y,r^*}^{(t)}, v_{u,q} \right\rangle$ is positive for $\Omega(k)$ tuples $(u, q) \in [k] \times [2]$ (under Setting B.14). We can similarly upper bound the update

to $\left\langle w_{y,r}^{(t)}, v_{s,\ell} \right\rangle$ for an arbitrary $r \in [m]$ as:

$$
\begin{aligned}
\left\langle -\eta \boldsymbol{\nabla}_{w_{y,r}^{(t)}} J_{MM}(g, \mathcal{X}), v_{s,\ell} \right\rangle \leq & -\frac{\eta}{N^2} \sum_{i \notin N_y} \sum_{j \in N_s} \phi^y\big(g_t(z_{i,j})\big) \Theta\left( \frac{\left\langle w_{y,r}^{(t)}, v_{s,\ell} \right\rangle^{\alpha-1}}{\rho^{\alpha-1}} \right) \\
& -\frac{\eta}{N^2} \sum_{i \notin N_y \cup N_s} \sum_{j \notin N_y \cup N_s} \phi^y\big(g_t(z_{i,j})\big) \Theta\left( \frac{P\delta_3^\alpha \min_{u \in [k], q \in [2]} \left\langle w_{y,r}^{(t)}, v_{u,q} \right\rangle^{\alpha-1}}{\rho^{\alpha-1}} \right) \\
& +\frac{\eta}{N^2} \sum_{i \in N_y} \sum_{j \in N_s} \big(1 - 2\phi^y\big(g_t(z_{i,j})\big)\big) \Theta\left( \frac{\max_{q \in [2]} \left\langle w_{y,r}^{(t)}, v_{s,\ell} + v_{y,q} \right\rangle^{\alpha-1}}{\rho^{\alpha-1}} \right) \\
& +\frac{\eta}{N^2} \sum_{i \in N_y} \sum_{j \notin N_y \cup N_s} \big(1 - 2\phi^y\big(g_t(z_{i,j})\big)\big) \Theta\left( \frac{\delta_4 \max_{u \in [k], q \in [2]} \left\langle w_{y,r}^{(t)}, v_{u,q} \right\rangle^{\alpha-1}}{\rho^{\alpha-1}} \right) \\
& +\frac{\eta}{N^2} \sum_{i \in N_y} \sum_{j \in N_y} \big(1 - \phi^y\big(g_t(z_{i,j})\big)\big) \Theta\left( \frac{P\delta_4^\alpha \max_{u \neq y, q \in [2]} \left\langle w_{y,r}^{(t)}, v_{u,q} \right\rangle^{\alpha-1}}{\rho^{\alpha-1}} \right)
\end{aligned}
$$
(B.20)

As the above may be rather difficult to parse on first glance, let us take a moment to unpack the individual terms on the RHS. The first two terms are a precise splitting of the $-2\phi^y(g_t)$ term from Calculation A.10; namely, the case where we mix with the class $s$ allows for constant size coefficients on the feature $v_{s,\ell}$ while the other cases only allow for $v_{s,\ell}$ to show up in the feature noise patches. The next three terms consider all cases of mixing with the class $y$. The first of these terms considers the case of mixing class $y$ with class $s$, in which case it is possible to have patches in $z_{i,j}$ that have both $v_{s,\ell}$ and $v_{y,\ell^*}$ with constant coefficients. The next term considers mixing class $y$ with a class that is neither $y$ nor $s$, in which case the feature $v_{s,\ell}$ can only show up when mixing with a feature noise patch, so we suffer a factor of at least $\delta_4 = \Theta(1/k^{1.5})$ (note we do not suffer a $\delta_4^\alpha$ factor as $v_{y,\ell^*}$ can still be in $z_{i,j}$) from the $\langle z_{i,j}, v_{s,\ell} \rangle$ part of the gradient. Finally, the last term considers mixing within class $y$.

The first of the three positive terms in the RHS of Equation B.20 presents the main problem - the fact that the diagonal correlations can show up non-trivially in the off-diagonal correlation gradient means the gradients can be much larger than in the ERM case. However, the key is that there are only $\Theta(N/k^2)$ mixed points between classes $y$ and class $s$. Thus, once more using the fact that $\Theta(|N_u|) = \Theta(N/k)$ for every $u \in [k]$, the other conditions in our setting, and Claim B.15, we obtain that for all $t \leq T_A$:

$$
\left\langle -\eta \boldsymbol{\nabla}_{w_{y,r^*}^{(t)}} J_{MM}(g, \mathcal{X}), v_{y,\ell^*} \right\rangle \geq \Theta\left( \frac{\eta \left\langle w_{y,r^*}^{(t)}, v_{y,\ell^*} \right\rangle^{\alpha-1}}{k\rho^{\alpha-1}} \right)
$$
(B.21)

$$
\left\langle -\eta \boldsymbol{\nabla}_{w_{y,r}^{(t)}} J_{MM}(g, \mathcal{X}), v_{s,\ell} \right\rangle \leq \Theta\left( \frac{\eta \max_{u \in \{s,y\}, q \in [2]} \left\langle w_{y,r}^{(t)}, v_{u,q} \right\rangle^{\alpha-1}}{k^2\rho^{\alpha-1}} \right)
$$
(B.22)

Crucially we have that Equation B.22 is a $\Theta(1/k)$ factor smaller than Equation B.21. Recalling that all correlations are $O(\log(k)/\sqrt{d})$ at initialization, we see that the difference in the updates in Equations B.21 and B.22 is at least of the same order as Equation B.21. Thus, in $O(\text{poly}(k))$ iterations, it follows that $\left\langle w_{y,r^*}^{(t)}, v_{y,\ell^*} \right\rangle > \left\langle w_{y,r}^{(t)}, v_{s,\ell} \right\rangle$ (this necessarily occurs for a $t < T_A$ by definition of $A$ and comparison to the bounds above), after which it follows from Equations B.21 and B.22 that $\left\langle w_{y,r}^{(T_A)}, v_{s,\ell} \right\rangle = O(\left\langle w_{y,r^*}^{(T_A)}, v_{y,\ell^*} \right\rangle / k)$ (and clearly $T_A = O(\text{poly}(k))$). This proves the first part of the claim.

It remains to show that the off-diagonal correlations also do not decrease by too much, as if they were to become negative that would potentially cause problems in Equation B.19 due to $\widetilde{\text{ReLU}}'$ becoming 0. Using Equation B.20, we can form the following lower bound to $\left\langle w_{y,r}^{(t)}, v_{s,\ell} \right\rangle$:

$$\left\langle -\eta \mathbf{\nabla}_{w_{y,r}^{(t)}} J_{MM}(g, \mathcal{X}), v_{s,\ell} \right\rangle \geq -\Theta \left( \frac{\eta \left\langle w_{y,r}^{(t)}, v_{s,\ell} \right\rangle^{\alpha-1}}{k^2 \rho^{\alpha-1}} \right) \tag{B.23}$$

Now let $T$ denote the number of iterations starting from initialization that it takes $\left\langle w_{y,r}^{(t)}, v_{s,\ell} \right\rangle$ to decrease to $\left\langle w_{y,r}^{(0)}, v_{s,\ell} \right\rangle / \text{polylog}(k)$ for some fixed $\text{polylog}(k)$ factor. Then it follows from Equation B.21 that in $T$ iterations $\left\langle w_{y,r^*}^{(t)}, v_{y,\ell^*} \right\rangle$ has increased by at least a $k/\text{polylog}(k)$ factor. As a result, we have that at $T_A$ the correlation $\left\langle w_{y,r}^{(t)}, v_{s,\ell} \right\rangle$ has decreased by at most a $\text{polylog}(k)^C$ factor for some universal constant $C$, proving the claim. ∎

**Corollary B.18.** For any class $y \in [k]$, and any $t \geq T_A$ (for any $T_A$ satisfying the definition in Claim B.17), we have for any $s \neq y$ and $\ell \in [2]$:

$$\left\langle w_{y,r}^{(t)}, v_{s,\ell} \right\rangle = O \left( \max_{\ell' \in [2]} \left\langle w_{y,r}^{(t)}, v_{y,\ell'} \right\rangle / k \right)$$

Additionally, for all $s, \ell$ with $\left\langle w_{y,r}^{(0)}, v_{s,\ell} \right\rangle > 0$, we have that $\left\langle w_{y,r}^{(t)}, v_{s,\ell} \right\rangle > 0$.

*Proof of Corollary B.18.* The $O(1/k)$ factor separation between the updates to diagonal and off-diagonal correlations shown in Equations B.21 and B.22 continue to hold once we pass into the linear regime of $\widetilde{\text{ReLU}}$. Furthermore, the logic used to prove the lower bound for positive correlations in Claim B.17 easily extends to showing that the correlations remain positive throughout training. ∎

As noted above, the bound on the off-diagonal correlations obtained in Claim B.17 and Corollary B.18 is much weaker than what it was in Claim B.4, which is why we weakened the assumptions in Claim B.15. We now prove the Midpoint Mixup analogues to Claims B.5, B.6, and B.7.

**Claim B.19.** Consider $y \in [k]$ and $t$ such that $\max_{i \in N_y, j \in [N]} g_t^y(z_{i,j}) = \Theta(\log k)$. Then $\max_{i \in N_y, j \in [N]} g_t^y(z_{i,j}) = \Theta(\min_{i \in N_y, j \in [N]} g_t^y(z_{i,j}))$.

*Proof of Claim B.19.* For any $t$ satisfying the conditions of the claim, we necessarily have that Corollary B.18 holds. As a result, we have:

$$\sum_{r \in [m]} \sum_{\ell \in [2]} \left\langle w_{y,r}^{(t)}, v_{y,\ell} \right\rangle = O(\log k) \implies \sum_{r \in [m]} \sum_{\ell \in [2]} \left\langle w_{y,r}^{(t)}, v_{s,\ell} \right\rangle = O(\log(k)/k)$$

Thus, we may disregard the off-diagonal correlations in considering the class $y$ output on $z_{i,j}$ (i.e. we do not need to worry about the $x_j$ part of $z_{i,j}$), and the rest is identical to Claim B.5. ∎

**Claim B.20.** For each $y \in [k]$, let $T_y$ denote the first iteration at which $\max_{i \in N_y, j \notin N_y} g_{T_y}^y(z_{i,j}) \geq \log(k-1)$. Then we have that $T_y = O(\text{poly}(k))$ (for a sufficiently large polynomial in $k$) and that $\min_{i \in N_y, j \notin [N]} g_t^y(z_{i,j}) = \Omega(\log k)$ for all $t \geq T_y$.

*Proof of Claim B.20.* As in the proof of Claim B.6, applying Claim B.17 to any class $y$ yields the existence of a correlation $\left\langle w_{y,r^*}^{(t)}, v_{y,\ell^*} \right\rangle$ and a $t = O(\text{poly}(k))$ such that $\left\langle w_{y,r^*}^{(t)}, v_{y,\ell^*} \right\rangle > \rho/(\delta_2 - \delta_1)$. And again, reusing the logic of Claim B.17 but replacing $\left\langle w_{y,r^*}^{(t)}, v_{y,\ell^*} \right\rangle$ in Equation B.21 with $\rho/(\delta_2 - \delta_1)$ yields that in an additional $O(\text{poly}(k))$ iterations we have $\left\langle w_{y,r^*}^{(t)}, v_{y,\ell^*} \right\rangle > 2\rho/\delta_1$ (implying that $w_{y,r^*}^{(t)}$ has reached the linear regime of $\widetilde{\text{ReLU}}$ on effectively all mixed points) while the off-diagonal correlations continue to lag behind by a $O(1/k)$ factor.

At this point we may lower bound the update to $\left\langle w_{y,r^*}^{(t)}, v_{y,\ell^*} \right\rangle$ as:

$$\left\langle -\eta \boldsymbol{\nabla}_{w_{y,r^*}^{(t)}} J_{MM}(g, \mathcal{X}), v_{y,\ell^*} \right\rangle \geq \frac{\eta}{N^2} \sum_{i \in N_y} \sum_{j \notin N_y} \Theta\Big(1 - 2\phi^y\big((g_t(z_{i,j}))\big)\Big)$$
$$+ \frac{\eta}{N^2} \sum_{i \in N_y} \sum_{j \in N_y} \Theta\Big(1 - \phi^y\big((g_t(z_{i,j}))\big)\Big)$$
$$- \frac{\eta}{N^2} \sum_{i \notin N_y} \sum_{j \notin N_y} \phi^y\big(g_t(z_{i,j})\big) \Theta\left(\frac{P\sigma_4^\alpha \max_{u \in [k], q \in [2]} \left\langle w_{y,r^*}^{(t)}, v_{u,q} \right\rangle^{\alpha-1}}{\rho^{\alpha-1}}\right) \tag{B.24}$$

Using Claim B.15, we have that so long as $\max_{i \in N_y,\ j \notin N_y} g_t^y(z_{i,j}) < \log(k-1)$, we get (using $|N_u| = \Theta(N/k)$ for all $u \in [k]$):

$$\left\langle -\eta \boldsymbol{\nabla}_{w_{y,r^*}^{(t)}} J_{MM}(g, \mathcal{X}), v_{y,\ell^*} \right\rangle \geq \Theta(\eta/k) \tag{B.25}$$

This also implies, by the logic of Claim B.17, that the off-diagonal correlations $\left\langle w_{y,r}^{(t)}, v_{s,\ell} \right\rangle$ have updates that can be upper bounded as:

$$\left\langle -\eta \boldsymbol{\nabla}_{w_{y,r}^{(t)}} J_{MM}(g, \mathcal{X}), v_{s,\ell} \right\rangle \leq \Theta(\eta/k^2) \tag{B.26}$$

Comparing Equations B.25 and B.26, we have that $g_t^y(z_{i,j}) \geq \log(k-1)$ (and, consequently, $g_t^y(x_i) \geq \log(k-1)$) for at least one mixed point $z_{i,j}$ with $i \in N_y$ in $O(\text{poly}(k))$ iterations while the off-diagonal correlations are $O(\log(k)/k)$. This also implies that $\min_{i \in N_y,\ j \in [N]} g_t^y(z_{i,j}) = \Omega(\log k)$ by Claim B.19. Finally, since Equation B.25 is positive, the class $y$ network outputs remain $\Omega(\log k)$ for $t \geq T_y$ (as again we cannot decrease below $\log(k-1)$ by more than $o(1)$ since the gradients are $o(1)$). ∎

**Part II.** Having analyzed the growth of diagonal and off-diagonal correlations in the initial stages of training, we now shift gears to focusing on the gaps between the correlations for each class. The key idea is that $J_{MM}$ will push the correlations for the features $v_{y,1}$ and $v_{y,2}$ closer together throughout training (so as long as they are sufficiently separated), for every class $y$.

In order to prove this, we will rely on analyzing an expectation form of the gradient for $J_{MM}$. As the expressions involved in this analysis can become cumbersome quite quickly, we will first introduce a slew of new notation to make the presentation of the results a bit easier.

Firstly, in everything that follows, we assume $v_{y,1}$ to be the better correlated feature at time $t$ for every class $y \in [k]$ in the following sense:

$$\sum_{r \in B_{y,1}^{(t)}} \left\langle w_{y,r}^{(t)}, v_{y,1} \right\rangle \geq \sum_{r \in B_{y,2}^{(t)}} \left\langle w_{y,r}^{(t)}, v_{y,2} \right\rangle \tag{B.27}$$

Where the sets $B_{y,\ell}^{(t)}$ are as defined in Equation B.16. Furthermore, as we will refer to the quantities in Equation B.27 many times, we use $C_{y,1}^{(t)}$ and $C_{y,2}^{(t)}$ to denote the LHS and RHS of Equation B.27, and $\Delta_y^{(t)} = C_{y,1}^{(t)} - C_{y,2}^{(t)}$.

Now for the aforementioned expectation analysis we introduce several relevant random variables. We use $\beta_{y,p}$ (for every $y \in [k]$) to denote a random variable following the distribution of the signal coefficients for class $y$ from Definition 4.1 and we further use $\beta_y$ to denote a random variable representing the sum of $C_P$ i.i.d. $\beta_{y,p}$. Similarly, we use $z_{y,s}$ to denote the average of two random variables following the distributions of class $y$ and class $s$ points respectively. Finally, we define $A_1(\beta_s, \beta_y)$ and $A_2(\beta_s, \beta_y)$ as:

$$A_1(\beta_s, \beta_y) \triangleq 1 - 2\phi^y\big((g_t(z_{y,s}))\big) \tag{B.28}$$
$$A_2(\beta_s, \beta_y) \triangleq 1 - \phi^y\big((g_t(z_{y,s}))\big) \tag{B.29}$$

In context, this notation will imply that $A_1(\beta_y, \beta_s) = 1 - 2\phi^s\big((g_t(z_{y,s}))\big)$ (i.e. swapping the order of arguments changes which coordinate of the softmax is being considered).

Now we will first prove an upper bound on the difference of gradient correlations in the linear regime, and then use these ideas to prove that correlations in the poly part of $\widetilde{\text{ReLU}}$ will still get significant gradient. After we have done that, we will revisit this next claim to show that the separation between feature correlations continues to decrease even after they reach the linear regime.

**Claim B.21.** Suppose that $\max_{s \in [k]} C_{s,1}^{(t)} = O(\log k)$. Then for any class $y \in [k]$ and any $r_1 \in B_{y,1}^{(t)}$ and $r_2 \in B_{y,2}^{(t)}$, we let:

$$\Psi(r_1, r_2) \triangleq \left\langle -\nabla_{w_{y,r_1}^{(t)}} J_{MM}(g, \mathcal{X}), v_{y,1} \right\rangle - \left\langle -\nabla_{w_{y,r_2}^{(t)}} J_{MM}(g, \mathcal{X}), v_{y,2} \right\rangle \quad (B.30)$$

After which we have that:

$$\Psi(r_1, r_2) \leq \Theta\left(\frac{1}{k^2}\right) \sum_{s \neq y} \mathbb{E}_{\beta_s, \beta_y}\left[A_1(\beta_s, \beta_y)(\beta_y - C_P\delta_2/2)\right]$$

$$+ \Theta\left(\frac{1}{k^2}\right) \mathbb{E}_{\beta_y}\left[A_2(\beta_y, \beta_y)(\beta_y - C_P\delta_2/2)\right]$$

$$+ O\left(P\delta_4^\alpha(\log k)^{\alpha-1}\right) \quad (B.31)$$

*Proof of Claim B.21.* Using the logic from Equation B.24 as well as the fact that $r_1 \in B_{y,1}^{(t)}$ and $r_2 \in B_{y,2}^{(t)}$ (i.e. we are considering weights in the linear regime of $\widetilde{\text{ReLU}}$ for each feature), we get:

$$\Psi(r_1, r_2) \leq \frac{1}{N^2} \sum_{i \in N_y} \sum_{j \notin N_y} \left(1 - 2\phi^y\big((g_t(z_{i,j}))\big)\right) \left(\sum_{p \in P_{y,1}(x_i)} \beta_{i,p} - C_P\delta_2/2\right)$$

$$+ \frac{1}{N^2} \sum_{i \in N_y} \sum_{j \notin N_y} \left(1 - \phi^y\big((g_t(z_{i,j}))\big)\right) \left(\sum_{p \in P_{y,1}(x_i)} \beta_{i,p} - C_P\delta_2/2\right)$$

$$+ O\left(P\delta_4^\alpha(\log k)^{\alpha-1}\right) \quad (B.32)$$

Now since we took $N$ sufficiently large in Assumption 4.3, by concentration for bounded random variables we can replace the expressions on the RHS above with their expected values, as the deviation will be within $O\left(P\delta_4^\alpha(\log k)^{\alpha-1}\right)$ (with probability $1 - O(1/k)$, consistent with our setting). However, the expectations will be over all of the random variables $\beta_u$ for $u \in [k]$, not just the classes $s$ and $y$ being mixed (or in the case of the second term above, just the class $y$).

Fortunately, we observe that for the mixed point random variable $z_{y,s}$, the $\beta_u$ for $u \neq y, s$ can only show up in the feature noise patches of $z_{y,s}$. Thus, by an identical calculation to the one controlling the feature noise contribution to the gradient above (once again, refer to Equation B.24), we see that we may consider the expectation over just $\beta_y$ and $\beta_s$ while marginalizing out the other random variables and staying within the error term above, thereby obtaining Equation B.31. $\blacksquare$

We will now show that $\mathbb{E}_{\beta_s, \beta_y}\left[A_1(\beta_s, \beta_y)(\beta_y - C_P\delta_2/2)\right]$ is significantly negative so long as the separation between feature correlations $\Delta_y^{(t)}$ is sufficiently large. Once again, to simplify notation even further, we will use $\tilde{\beta}_y = \beta_y - C_P\delta_2/2$ and use $\mathbb{P}(\tilde{\beta}_y)$ to refer to its associated probability measure. Furthermore, we will use:

$$D_{y,s}^{(t)} = (C_{y,1}^{(t)} + C_{y,2}^{(t)}) - (C_{s,1}^{(t)} + C_{s,2}^{(t)})$$

In other words, $D_{y,s}^{(t)}$ represents the difference in the linear outputs of classes $y$ and $s$. With this in mind, we can prove the aforementioned result.

**Claim B.22.** Suppose that $\max_{u \in [k]} C_{u,1}^{(t)} = O(\log k)$. Let $y$ be any class such that $\Delta_y^{(t)} \geq \log k - o(1)$, and suppose that there exists at least one class $s \in [k]$ such that there is a set $U \subset [0, C_P(\delta_2/2-$

$\delta_1)] \times [-C_P(\delta_2/2 - \delta_1), C_P(\delta_2/2 - \delta_1)]$ with $(\mathbb{P}(\tilde{\beta}_y) \times \mathbb{P}(\tilde{\beta}_s))(U) \geq 0.01$ (i.e. its measure is at least 0.01) and for all $(a, b) \in U$ we have:

$$(b - C_P\delta_2/2)\Delta_s^{(t)} - C_P\delta_2 D_{y,s}^{(t)}/2 \leq (a - C_P\delta_2/2)\Delta_y^{(t)} \tag{B.33}$$

Then we have:

$$\mathbb{E}_{\beta_s,\beta_y}\left[A_1(\beta_s, \beta_y)(\beta_y - C_P\delta_2/2)\right] = -\Theta(1) \tag{B.34}$$

*Proof of Claim B.22.* We begin by first showing that the expectation on the LHS of Equation B.34 is negative. Indeed, this is almost immediate from the fact that $\tilde{\beta}_y$ is a symmetric, mean zero random variable - we need only show that $A_1$ is monotonically decreasing in $\beta_y$.

From the definition of $A_1$, we observe that it suffices to show that $g_t^y(z_{y,s})$ is monotonically increasing in $\beta_y$. However, this is straightforward to see from the assumption that $\Delta_y^{(t)} \geq \log k - o(1)$, as this implies that an $\epsilon$ increase in $\beta_y$ leads to a $O(\epsilon \log k) - O(\epsilon)$ increase in $g_t^y$, since the feature noise and weights that are in the polynomial part of $\widetilde{\text{ReLU}}$ can contribute at most $O(1)$ by the logic of Claim B.19.

Now we need only show that the expectation is sufficiently negative. To do this, we will rely on the following facts, which will allow us to write things purely in terms of $C_{y,\ell}^{(t)}$ and $C_{s,\ell}^{(t)}$ (i.e. disregarding the weights that are not in the linear regime):

$$g_t^y(z_{y,s}) \in \left[\beta_y C_{y,1}^{(t)} + (C_P\delta_2 - \beta_y)C_{y,2}^{(t)}, \ \beta_y C_{y,1}^{(t)} + (C_P\delta_2 - \beta_y)C_{y,2}^{(t)} + O(1)\right] \tag{B.35}$$

$$g_t^s(z_{y,s}) \in \left[\beta_s C_{s,1}^{(t)} + (C_P\delta_2 - \beta_s)C_{s,2}^{(t)}, \ \beta_s C_{s,1}^{(t)} + (C_P\delta_2 - \beta_s)C_{s,2}^{(t)} + O(1)\right] \tag{B.36}$$

$$g_t^u(z_{y,s}) = O\left(\frac{\log k}{k}\right) \quad \text{for } u \neq y, s \tag{B.37}$$

Which follow from Claim B.19 and Corollary B.18 (alongside the assumption that $\max_{u \in [k]} C_{u,1}^{(t)} = O(\log k)$) respectively. Now we perform the substitution $g_t^u \leftarrow g_t^u - C_P\delta_2(C_{y,1}^{(t)} + C_{y,2}^{(t)})/2$ for all $u \in [k]$, as this can be done without changing the value of $\phi^y(g_t(z_{y,s}))$. Under this transformation we have that (using Equation B.35):

$$g_t^y(z_{y,s}) \in \left[(\beta_y - C_P\delta_2/2)\Delta_y^{(t)}, (\beta_y - C_P\delta_2/2)\Delta_y^{(t)} + O(1)\right] \tag{B.38}$$

$$g_t^s(z_{y,s}) \in \left[(\beta_s - C_P\delta_2/2)\Delta_s^{(t)} - C_P\delta_2 D_{y,s}^{(t)}/2, (\beta_s - C_P\delta_2/2)\Delta_s^{(t)} - C_P\delta_2 D_{y,s}^{(t)}/2 + O(1)\right] \tag{B.39}$$

Which isolates the correlation gap term $\Delta_y^{(t)}$. Prior to proceeding further we will let $\Lambda_s^{(t)} \triangleq \sum_{u \neq y} \exp(g_t^u(z_{y,s}))$, so as to prevent the equations to follow from becoming too unwieldy. Now we have:

$$\mathbb{E}_{\beta_s,\beta_y}\left[A_1(\beta_s, \beta_y)(\beta_y - C_P\delta_2/2)\right]$$

$$= \int_{C_P\delta_1}^{C_P(\delta_2-\delta_1)} \int_{C_P\delta_1}^{C_P(\delta_2-\delta_1)} \frac{\Lambda_s^{(t)} - \exp(g_t^y(z_{y,s}))}{\Lambda_s^{(t)} + \exp(g_t^y(z_{y,s}))}(\beta_y - C_P\delta_2/2) \, d\mathbb{P}(\beta_y)d\mathbb{P}(\beta_s)$$

$$\leq \int_{C_P(\delta_1-\delta_2/2)}^{C_P(\delta_2/2-\delta_1)} \int_{C_P(\delta_1-\delta_2/2)}^{0} \frac{\Lambda_s^{(t)} - \exp\left(\tilde{\beta}_y\Delta_y^{(t)} + O(1)\right)}{\Lambda_s^{(t)} + \exp\left(\tilde{\beta}_y\Delta_y^{(t)} + O(1)\right)}\tilde{\beta}_y \, d\mathbb{P}(\tilde{\beta}_y)d\mathbb{P}(\tilde{\beta}_s)$$

$$+ \int_{C_P(\delta_1-\delta_2/2)}^{C_P(\delta_2/2-\delta_1)} \int_{0}^{C_P(\delta_2/2-\delta_1)} \frac{\Lambda_s^{(t)} - \exp\left(\tilde{\beta}_y\Delta_y^{(t)}\right)}{\Lambda_s^{(t)} + \exp\left(\tilde{\beta}_y\Delta_y^{(t)}\right)}\tilde{\beta}_y \, d\mathbb{P}(\tilde{\beta}_y)d\mathbb{P}(\tilde{\beta}_s) \tag{B.40}$$

Where above we used Equation B.38 to get the upper bound in the last step. We next focus on bounding the inner integral in Equation B.40. Using the symmetry of $\tilde{\beta}_y$, we have that:

$$
\mathbb{E}_{\beta_y}\left[A_1(\beta_s, \beta_y)(\beta_y - C_P\delta_2/2)\right]
$$

$$
= -\int_0^{C_P(\delta_2/2-\delta_1)} \left(\frac{\Lambda_s^{(t)} - \exp\left(-\tilde{\beta}_y\Delta_y^{(t)} + O(1)\right)}{\Lambda_s^{(t)} + \exp\left(-\tilde{\beta}_y\Delta_y^{(t)} + O(1)\right)} - \frac{\Lambda_s^{(t)} - \exp\left(\tilde{\beta}_y\Delta_y^{(t)}\right)}{\Lambda_s^{(t)} + \exp\left(\tilde{\beta}_y\Delta_y^{(t)}\right)}\right) \tilde{\beta}_y \; d\mathbb{P}(\tilde{\beta}_y)
$$

$$
= -\int_0^{C_P(\delta_2/2-\delta_1)} \frac{2\Lambda_s^{(t)}\left(\exp\left(\tilde{\beta}_y\Delta_y^{(t)}\right) - \exp\left(-\tilde{\beta}_y\Delta_y^{(t)} + O(1)\right)\right)}{(\Lambda_s^{(t)})^2 + \Lambda_s^{(t)}\left(\exp\left(\tilde{\beta}_y\Delta_y^{(t)}\right) + \exp\left(-\tilde{\beta}_y\Delta_y^{(t)} + O(1)\right)\right) + \exp(O(1))} \tilde{\beta}_y \; d\mathbb{P}(\tilde{\beta}_y)
$$

$$(\text{B.41})$$

One can sanity check that Equation B.41 is bounded below by -1, as we would expect. The only tricky aspect of Equation B.41 is the $O(1)$ term in the exponential, which can lead to a positive contribution (via a negative integrand) when $\tilde{\beta}_y$ is close to 0. However, we can safely restrict the bounds of integration in Equation B.41 to a region $[\varrho, C_P(\delta_2/2 - \delta_1)]$ for $\varrho = \Theta(1/\log k)$ (with an appropriately chosen constant), as in such a region the integrand is guaranteed to be positive since $\Delta_y^{(t)} \geq \log k - o(1)$. Furthermore, this restriction does not cost us anything (like an additional positive term), as concern from Equation B.41 is purely a consequence of how we bounded $g_t^y$ and $g_t^s$. Indeed, by our earlier monotonicity argument it is clear that we can cut out the region corresponding to $[-\varrho, \varrho]$ from the first line of Equation B.40 without decreasing the RHS.

Furthermore, we also have from $\Delta_y^{(t)} \geq \log k - o(1)$ and Equation B.37 that $\sum_{u \neq y,s} \exp(g_t^u(z_{y,s})) = O(1/k^{C_P\delta_2 - 1})$ (after making the adjustment $g_t^u \leftarrow g_t^u - C_P\delta_2(C_{y,1}^{(t)} + C_{y,2}^{(t)})/2$ that we did above). Now using our assumption in the statement of the claim that Equation B.33 holds for some set $U$, we obtain:

$$
\mathbb{E}_{\beta_s, \beta_y}\left[A_1(\beta_s, \beta_y)(\beta_y - C_P\delta_2/2)\right]
$$

$$
\leq -\int_{C_P(\delta_1-\delta_2/2)}^{C_P(\delta_2/2-\delta_1)} \int_\varrho^{C_P(\delta_2/2-\delta_1)} \Theta\left(\frac{\tilde{\beta}_y \exp\left(\tilde{\beta}_y\Delta_y^{(t)}\right)}{O(k^{1-C_P\delta_2}) + \exp\left(\tilde{\beta}_y\Delta_y^{(t)}\right)}\right) d\mathbb{P}(\tilde{\beta}_y)d\mathbb{P}(\tilde{\beta}_s)
$$

$$
= -\Theta(1) \tag{B.42}
$$

Where the last line follows after restricting the bounds of integration of the two integrals to their intersections with $U$ (this allows us to disregard $g_t^s$ in the asymptotic expression above via Equation B.39). This proves the claim. ∎

We proved Claim B.22 in terms of $\beta_y$ (the sum of the individual $\beta_{y,p}$) to keep notation manageable (avoids $C_P$ iterated integrals) and to more closely mirror the proof of Proposition 3.4. However, what we will really use for our remaining analysis is the following corollary, which gives the same result as Claim B.22 but for each of the individual terms $\beta_{y,p}$. Below we use $\sum_{i=1}^{C_P} \beta_{y,i}$ to make explicit the dependence between the sum and each individual random variable $\beta_{y,p}$ (so as to not mislead one to think of them as independent random variables).

**Corollary B.23.** Under the same conditions as Claim B.22, we have for every $p \in [C_P]$:

$$
\mathbb{E}_{\beta_s, \beta_{y,1}, \ldots, \beta_{y,C_P}}\left[A_1\left(\beta_s, \sum_{i=1}^{C_P} \beta_{y,i}\right)(\beta_{y,p} - \delta_2/2)\right] = -\Theta(1) \tag{B.43}
$$

*Proof of Claim B.23.* The proof follows identically to that of Claim B.23 (effectively the only change in the computations is that $\tilde{\beta}_y$ becomes $\tilde{\beta}_{y,p}$), as the functions $A_1$ and $A_2$ satisfy the same monotonically increasing property in each of the i.i.d. $\beta_{y,p}$ (and there are only $C_P = \Theta(1)$ many). One could also have simply seen this from the symmetry of the $\beta_{y,p}$ in Equation B.43; indeed, we expect Equation B.43 to only differ by a factor $1/C_P$ from Equation B.34. ∎

Now we can show using Corollary B.23 that there is a significant gradient component towards correcting the separation between the feature correlations even when the second feature correlation is in the polynomial part of $\widetilde{\text{ReLU}}$ (which is where it got stuck for a significant number of classes in the ERM proof).

**Claim B.24.** Suppose that $\max_{u \in [k]} C_{u,1}^{(t)} = O(\log k)$. Let $y$ be any class such that $\Delta_y^{(t)} \geq \log k - o(1)$. Then for any $r_2 \notin B_{y,2}^{(t)}$ satisfying $\left\langle w_{y,r_2}^{(0)}, v_{y,2} \right\rangle \geq \tau > 0$, so long as there exists an $r_1 \in B_{y,1}^{(t)}$ satisfying:

$$\left\langle -\boldsymbol{\nabla}_{w_{y,r_1}^{(t)}} J_{MM}(g, \mathcal{X}), v_{y,1} \right\rangle \geq 0 \tag{B.44}$$

We have:

$$\left\langle -\boldsymbol{\nabla}_{w_{y,r_2}^{(t)}} J_{MM}(g, \mathcal{X}), v_{y,2} \right\rangle \geq \Theta\left(\frac{\tau}{\rho^{\alpha-1}k^2}\right) \tag{B.45}$$

*Proof of Claim B.24.* By near-identical logic to the steps leading to Equation B.31 and using Equation B.21, we obtain (following the same notation as before):

$$\left\langle -\boldsymbol{\nabla}_{w_{y,r_2}^{(t)}} J_{MM}(g, \mathcal{X}), v_{y,2} \right\rangle \geq \Theta\left(\frac{\tau}{\rho^\alpha k^2}\right) \sum_{s \neq y} \mathbb{E}_{\beta_s, \beta_{y,1}, \dots, \beta_{y,C_P}} \left[ A_1(\beta_s, \sum_{p=1}^{C_P} \beta_{y,p}) \sum_{p=1}^{C_P} (\delta_2 - \beta_{y,p})^\alpha \right] \tag{B.46}$$

Where above we have absorbed a factor of $1/2$ resulting from mixing classes and the feature noise component into the asymptotic term in front of the summation.

Now we break the rest of the proof into two cases: whether Equation B.33 holds or not. In the former case, using Corollary B.23 and our assumption in the statement of the claim that Equation B.44 holds, we get (from linearity of expectation):

$$\Theta\left(\frac{1}{k^2}\right) \sum_{s \neq y} \mathbb{E}_{\beta_s, \beta_{y,1}, \dots, \beta_{y,C_P}} \left[ A_1(\beta_s, \sum_{p=1}^{C_P} \beta_{y,p})(\delta_2 - \beta_{y,p}) \right] \geq \Theta\left(\frac{1}{k^2}\right) \tag{B.47}$$

Now observing that $\text{Cov}(A_1(\beta_s, \sum_{p=1}^{C_P} \beta_{y,p})(\delta_2 - \beta_{y,p}), (\delta_2 - \beta_{y,p})^{\alpha-1}) > 0$ for every $p$, we obtain:

$$\left\langle -\boldsymbol{\nabla}_{w_{y,r_2}^{(t)}} J_{MM}(g, \mathcal{X}), v_{y,2} \right\rangle$$
$$\geq \sum_{s \neq y} \sum_{p=1}^{C_P} \mathbb{E}_{\beta_s, \beta_{y,1}, \dots, \beta_{y,C_P}} \left[ A_1(\beta_s, \sum_{p=1}^{C_P} \beta_{y,p})(\delta_2 - \beta_{y,p}) \right] \mathbb{E}_{\beta_s, \beta_{y,1}, \dots, \beta_{y,C_P}}[(\delta_2 - \beta_{y,p})^{\alpha-1}] \tag{B.48}$$

And the result then follows for this case from Equation B.47 and the fact that $\mathbb{E}_{\beta_s, \beta_{y,1}, \dots, \beta_{y,C_P}}[(\delta_2 - \beta_{y,p})^{\alpha-1}]$ is a data-distribution-dependent constant.

For the second case when Equation B.33 does not hold, we have that *for every* class $s \neq y$ there exists a set $U' \subset [0, C_P(\delta_2/2 - \delta_1)] \times [-C_P(\delta_2/2 - \delta_1), C_P(\delta_2/2 - \delta_1)]$ such that $(\mathbb{P}(\tilde{\beta}_y) \times \mathbb{P}(\tilde{\beta}_s))(U') \geq 0.49$ (note the total measure of the set which $U'$ is a subset of is 0.5, by symmetry) and for all $(a, b) \in U'$ we have:

$$(b - C_P\delta_2/2)\Delta_s^{(t)} - C_P\delta_2 D_{y,s}^{(t)}/2 > (a - C_P\delta_2/2)\Delta_y^{(t)} \tag{B.49}$$

By our anti-concentration assumption on the $\beta_{y,p}$ it immediately follows that $D_{y,s}^{(t)} = -\Theta(\log k)$, from which we obtain that the the expectation terms in Equation B.46 are all $\Theta(1)$, so we are done. $\blacksquare$

Having proved Claim B.24, it remains to prove that both Equation B.44 and $\max_{y \in [k]} C_{y,1}^{(t)} = O(\log k)$ hold throughout training, as after doing so we can conclude that the second feature correlation will escape the polynomial part of $\widetilde{\text{ReLU}}$ and become sufficiently large in polynomially many training steps.

**Claim B.25.** For any $y \in [k], \ell \in [2]$, and $r \in [m]$, we have that:

$$\left\langle -\mathbf{\nabla}_{w_{y,r}^{(t+1)}} J_{MM}(g, \mathcal{X}), v_{y,\ell} \right\rangle \geq 0.99 \left\langle -\mathbf{\nabla}_{w_{y,r}^{(t)}} J_{MM}(g, \mathcal{X}), v_{y,\ell} \right\rangle$$

So long as $\sum_{s \neq y} \exp\left(g_{t+1}^s(z_{i,j})\right) \geq \sum_{s \neq y} \exp(g_t^s(z_{i,j}))$ for all mixed points $z_{i,j}$ with $i \in N_y$.

*Proof of Claim B.25.* We proceed by brute force; namely, as long as $\eta$ is sufficiently small, we can prove that the gradient for $J_{MM}$ does not decrease too much between successive iterations. As notation is going to become cumbersome quite quickly, we will use the following place-holders for the gradient correlations at time $t$ and $t + 1$:

$$G_t \triangleq \left\langle -\mathbf{\nabla}_{w_{y,r}^{(t)}} J_{MM}(g, \mathcal{X}), v_{y,\ell} \right\rangle$$

We will now prove the result assuming $r \in B_{y,\ell}^{(t)}$, as the the case where $r \notin B_{y,\ell}^{(t)}$ is strictly better (we will have the upper bound shown below with additional $o(1)$ factors). We have that (compare to Equation B.24):

$$\begin{aligned}
G_t - G_{t+1} \leq & \frac{1}{N^2} \sum_{i \in N_y} \sum_{j \notin N_y} \Theta\Big( \phi^y\big( (g_{t+1}(z_{i,j})) \big) - \phi^y\big( (g_t(z_{i,j})) \big) \Big) \\
& + \frac{1}{N^2} \sum_{i \in N_y} \sum_{j \in N_y} \Theta\Big( \phi^y\big( (g_{t+1}(z_{i,j})) \big) - \phi^y\big( (g_t(z_{i,j})) \big) \Big) \\
& + \frac{1}{N^2} \sum_{i \notin N_y} \sum_{j \notin N_y} \Big( \phi^y\big( (g_{t+1}(z_{i,j})) \big) - \phi^y\big( (g_t(z_{i,j})) \big) \Big) \Theta\left( \frac{P\sigma_4^\alpha \max_{u \in [k], q \in [2]} \left\langle w_{y,r^*}^{(t)}, v_{u,q} \right\rangle^{\alpha-1}}{\rho^{\alpha-1}} \right)
\end{aligned}$$

$$\text{(B.50)}$$

Let us now focus on the $\phi^y\big( (g_{t+1}(z_{i,j})) \big) - \phi^y\big( (g_t(z_{i,j})) \big)$ terms present in Equation B.50 above. We will just consider the first case above (mixing between class $y$ and a non-$y$ class), as the other analyses follow similarly. Furthermore, we will omit the $z_{i,j}$ in what follows (in the interest of brevity) and simply write $g_{t+1}^y$. Additionally, similar to Claim B.22, we will use the notation $\Lambda_t = \sum_{s \neq y} \exp(g_t^s)$.

Now by the assumption in the statement of the claim, we have that $\Lambda_{t+1} \geq \Lambda_t$, and since $m = \Theta(k)$ (number of weights per class), we have that $g_{t+1}^y \leq g_t^y + \Theta(k\eta G_t)$ (all of the updates for weights in the linear regime are identical and strictly larger than updates for those in the poly regime). Thus,

$$\begin{aligned}
\phi^y(g_{t+1}) - \phi^y(g_t) & \leq \frac{\exp(g_t^y + \Theta(k\eta G_t))}{\Lambda_t + \exp(g_t^y + \Theta(k\eta G_t))} - \frac{\exp(g_t^y)}{\Lambda_t + \exp(g_t^y)} \\
& = \frac{\Lambda_t \exp(g_t^y)(\exp(\Theta(k\eta G_t)) - 1)}{\Lambda_t^2 + \Lambda_t \exp(g_t^y)(\exp(\Theta(k\eta G_t)) + 1) + \exp(2g_t^y + \Theta(k\eta G_t))} \\
& \leq \Theta(k\eta G_t + k^2\eta^2 G_t^2) = \Theta(k\eta G_t) \qquad\qquad \text{(B.51)}
\end{aligned}$$

Where in the last line we again used the inequality $\exp(x) \leq 1 + x + x^2$ for $x \in [0, 1]$. Plugging Equation B.51 into Equation B.50 (after similar calculations for the other two pieces of Equation B.50) yields the result (since again, $\eta = O(1/\text{poly}(k))$ is suitably small). ∎

**Corollary B.26.** For any $y \in [k], \ell \in [2], r \in [m]$, and $t$, we have that:

$$\left\langle -\mathbf{\nabla}_{w_{y,r}^{(0)}} J_{MM}(g, \mathcal{X}), v_{y,\ell} \right\rangle \geq 0 \implies \left\langle -\mathbf{\nabla}_{w_{y,r}^{(t)}} J_{MM}(g, \mathcal{X}), v_{y,\ell} \right\rangle \geq 0$$

*Proof of Corollary B.26.* For every class $y$, we have $\sum_{s \neq y} \exp\left(g_{t+1}^s(z_{i,j})\right) \geq \sum_{s \neq y} \exp(g_t^s(z_{i,j}))$ for all mixed points $z_{i,j}$ with $i \in N_y$ for $t = 0$ (see the proof of Claim B.17), and the corollary then follows from an induction argument and Claim B.25. ∎

And with Corollary B.26 we may prove that $\max_{y \in [k]} C_{y,1}^{(t)} = O(\log k)$ holds throughout training.

**Claim B.27.** For all $t = O(\text{poly}(k))$, for any polynomial in $k$, we have that $\max_{y \in [k]} C_{y,1}^{(t)} = O(\log k)$.

*Proof of Claim B.27.* The idea is to consider the sum of gradient correlations across classes, and show that the cross-class mixing term in this sum becomes smaller (as this would be our only concern - we already know the same-class mixing term will become smaller by the logic of Claim B.6).

As in the previous claims in this section, we will proceed with an expectation analysis. We will focus on the weights $w_{y,r}$ that are in the linear regime for the feature $v_{y,1}$ for each class $y$, as these are the only relevant weights for $C_{y,1}^{(t)}$. Additionally, instead of considering the sum of gradient correlations over all $w_{y,r}$ with $r \in B_{y,1}^{(t)}$, it will suffice for our purposes to just consider the sum of gradient correlations over classes while using an arbitrary weight $w_{y,r}$ in the linear regime. Thus, we will abuse notation slightly and use $w_{y,r}$ to indicate such a weight for each class $y$ for the remainder of the proof of this claim (note that we do not mean to imply by this that weight $r$ is in the linear regime for every class simultaneously, but rather that there exists some $r$ for every class that is in the linear regime).

Now in the same vein as Equation B.31 (referring again to Equation B.24), we have that:

$$\sum_{y \in [k]} \left\langle -\nabla_{w_{y,r}^{(t)}} J_{MM}(g, \mathcal{X}), v_{y,1} \right\rangle \leq \Theta\left(\frac{1}{k^2}\right) \sum_{y=1}^{k} \sum_{s=y+1}^{k} \mathbb{E}_{\beta_s, \beta_y}[A_1(\beta_s, \beta_y)\beta_y] + \mathbb{E}_{\beta_y, \beta_s}[A_1(\beta_y, \beta_s)\beta_s]$$

$$+ \Theta\left(\frac{1}{k^2}\right) \sum_{y=1}^{k} \mathbb{E}_{\beta_y}[A_2(\beta_y, \beta_y)\beta_y] - O\left(P\delta_4^\alpha/\text{poly}(k)\right)$$

(B.52)

And we recall that $N$ is sufficiently large so that the deviations from the expectations above are negligible compared to the subtracted term. We have carefully paired the expectations in the leading term of Equation B.52 so as to make use of the following fact:

$$A_1(\beta_s, \beta_y) = -A_1(\beta_y, \beta_s) + \frac{\sum_{u \in [k] \setminus \{y,s\}} \exp(g_t^u(z_{y,s}))}{\sum_{u \in [k]} \exp(g_t^u(z_{y,s}))}$$

(B.53)

The second term on the RHS of Equation B.53 is of course $o(P\delta_4^\alpha/\text{poly}(k))$ so long as $g_t^s(z_{y,s})$ and/or $g_t^y(z_{y,s})$ are greater than $C \log k$ for a large enough constant $C$, so we obtain:

$$\sum_{y \in [k]} \left\langle -\nabla_{w_{y,r}^{(t)}} J_{MM}(g, \mathcal{X}), v_{y,1} \right\rangle \leq \Theta\left(\frac{1}{k^2}\right) \sum_{y=1}^{k} \sum_{s=y+1}^{k} \mathbb{E}_{\beta_s, \beta_y}[A_1(\beta_s, \beta_y)(\beta_y - \beta_s)]$$

$$+ \Theta\left(\frac{1}{k^2}\right) \sum_{y=1}^{k} \mathbb{E}_{\beta_y}[A_2(\beta_y, \beta_y)\beta_y] - O\left(P\delta_4^\alpha/\text{poly}(k)\right)$$

(B.54)

Now again by the logic of Claim B.22 we have that $\text{Cov}(A_1(\beta_s, \beta_y), \beta_y - \beta_s) < 0$, so it follows that:

$$\sum_{y \in [k]} \left\langle -\nabla_{w_{y,r}^{(t)}} J_{MM}(g, \mathcal{X}), v_{y,1} \right\rangle \leq \Theta\left(\frac{1}{k^2}\right) \sum_{y=1}^{k} \mathbb{E}_{\beta_y}[A_2(\beta_y, \beta_y)\beta_y] - O\left(P\delta_4^\alpha/\text{poly}(k)\right)$$

(B.55)

And if $g_t^y(z_{y,s}) \geq C \log k$ for a sufficiently large constant $C$, we have that the RHS above would be negative, which contradicts Corollary B.26, proving the claim. ∎

We have now wrapped up all of the pieces necessary to prove Theorem 4.7. Indeed, we can now show that for every class the correlation with both features becomes large over the course of training.

**Claim B.28.** For every class $y \in [k]$, in $O(\text{poly}(k))$ iterations (for a sufficiently large polynomial in $k$) we have that both $C_{y,1}^{(t)} = \Omega(\log k)$ and $C_{y,2}^{(t)} = \Omega(\log k)$.

*Proof of Claim B.28.* Claim B.20 guarantees $C_{y,1}^{(t)} = \Omega(\log k)$ in polynomially many iterations. If at this point $C_{y,2}^{(t)} = \Omega(\log k)$, we are done. If this is not the case, but we have $B_{y,2}^{(t)} \neq \emptyset$, then we are done by Claim B.21, Corollary B.26, and Claim B.27. On the other hand, if $B_{y,2}^{(t)} = \emptyset$, then we have by Claim B.24 that in polynomially many iterations (as we can take $\tau = \Theta(1/\sqrt{d})$ by our setting) $B_{y,2}^{(t)} \neq \emptyset$, after which we have reverted back to the previous case and we are still done. ∎

Now we may conclude the overall proof by observing that Claims B.20 and B.27 in tandem imply that we achieve (and maintain) perfect training accuracy in polynomially many iterations, while Claim B.28 implies that both features are learned in the sense of Definition 4.5. □

## C  PROOFS OF AUXILIARY RESULTS

### C.1  PROOFS OF LEMMA 3.2 AND PROPOSITION 3.3

**Lemma 3.2.** [Midpoint Mixup Optimal Direction] A linear model $g$ satisfies the following:

$$\lim_{\gamma \to \infty} J_{MM}(\gamma g, \mathcal{X}) = \inf J_{MM}(h, \mathcal{X}) \tag{3.3}$$

If $g$ has the property that for every class $y$ we have $\langle w_y, v_{y,\ell_1} \rangle = \langle w_s, v_{s,\ell_2} \rangle > 0$ and $\langle w_y, v_{s,\ell_2} \rangle \leq 0$ for every $s \neq y$ and $\ell_1, \ell_2 \in [2]$. Furthermore, with probability $1 - \exp(-\Theta(N))$ (over the randomness of $\mathcal{X}$), the condition $\langle w_y, v_{y,\ell_1} \rangle = \langle w_s, v_{s,\ell_2} \rangle$ is necessary for $g$ to satisfy Equation 3.3.

*Proof.* We first prove sufficiency. If $g$ satisfies the conditions in the Lemma, then we have for any data point $(x_i, y_i)$ that $g^{y_i}(x_i) = \langle w_{y_i}, v_{y_i,1} + v_{y_i,2} \rangle > 0$. We also have that $g^s(x_i) = 0$ for any $s \neq y_i$ (by the cross-class orthogonality condition). Letting $C = \langle w_y, v_{y,1} + v_{y,2} \rangle$ (note that this correlation is the same independent of $y$ due to the conditions of the lemma), we then get:

$$\phi^y(\gamma g(z_{i,j})) = \phi^y(\gamma g^{y_j}(z_{i,j})) = \frac{\exp(\gamma C/2)}{O(k-2) + 2\exp(\gamma C/2)} \tag{C.1}$$

For any mixed point $z_{i,j}$ with $y_i \neq y_j$. Equation C.1 tends to $1/2$ as $\gamma \to \infty$, and one can easily check that this is the global optimal prediction for the classes $y_i$ and $y_j$ on the Mixup point $z_{i,j}$ (for any such mixed point). Similarly, if $z_{i,j}$ is a mixed point with $y_i = y_j$, then Equation C.1 becomes the ERM case, we obtain the optimal prediction of 1 for the correct class in the limit.

On the other hand, if there exists a pair of classes $(y, s)$ with $s \neq y$ and $\ell_1, \ell_2 \in [2]$ such that $\langle w_y, v_{y,\ell_1} \rangle \neq \langle w_s, v_{s,\ell_2} \rangle$, then with probability $1 - \exp(-\Theta(N))$ there exists a mixed point $z_{i,j}$ in $\mathcal{X}$ (where $y_i = y$, $y_j = s$, and $y \neq s$) such that $g^y(z_{i,j}) \neq g^s(z_{i,j})$, and hence $\lim_{\gamma \to \infty} \phi^y(\gamma g^{(}z_{i,j})) \neq 1/2$, so we cannot achieve the infimum of the Midpoint Mixup loss. □

**Proposition 3.3.** For any distribution $\mathcal{D}_\lambda$ that is not a point mass on $0, 1$, or $1/2$, and any linear model $g$ satisfying the conditions of Lemma 3.2, we have that with probability $1 - \exp(-\Theta(N))$ (over the randomness of $\mathcal{X}$) there exists an $\epsilon_0 > 0$ depending only on $\mathcal{D}_\lambda$ such that:

$$J_M(g, \mathcal{X}, \mathcal{D}_\lambda) \geq \inf J_M(h, \mathcal{X}, \mathcal{D}_\lambda) + \epsilon_0 \tag{3.4}$$

*Proof.* Firstly, we observe (just from properties of cross-entropy):

$$\inf J_M(h, \mathcal{X}, \mathcal{D}_\lambda) = -\mathbb{E}_{\lambda \sim \mathcal{D}_\lambda}[\lambda \log \lambda + (1 - \lambda) \log(1 - \lambda)] \tag{C.2}$$

Now suppose a model $g$ satisfies the conditions of Lemma 3.2. Then we have that $g^{y_i}(x_i) = g^{y_j}(x_j) = C > 0$ for some constant $C$ and every pair $(x_i, y_i)$ and $(x_j, y_j)$.

As before, with at least probability $1 - \exp(-\Theta(N))$, we have that there exist a pair of points $(x_i, y_i)$ and $(x_j, y_j)$ in $\mathcal{X}$ with $y_i \neq y_j$. The Mixup loss restricted to this pair (for which we use the notation $J_M(g, z_{i,j}, \mathcal{D}_\lambda)$) is then:

$$J_M(g, z_{i,j}, \mathcal{D}_\lambda) = -\mathbb{E}_{\lambda \sim \mathcal{D}_\lambda}\left[\lambda \log \phi^{y_i}(g(z_{i,j})) + (1 - \lambda) \log \phi^{y_j}(g(z_{i,j}))\right] \tag{C.3}$$

Furthermore, we have that:

$$\phi^{y_i}(\gamma g(z_{i,j}(\lambda))) = \frac{\exp(\lambda C)}{O(k-2) + \exp(\lambda C) + \exp((1-\lambda)C)} \tag{C.4}$$

From Equations C.3 and C.4 we can see that, since $D_\lambda$ is supported on more than just $0, 1,$ and $1/2$, $J_M(g, z_{i,j}, D_\lambda) \to \infty$ as $C \to \infty$ (Equation C.4 implies that in the limit $\phi^{y_i}(\gamma g(z_{i,j}(\lambda)))$ can only take the values, $0, 1,$ or $1/2$). It is also easy to see that the same behavior occurs if one considers $C \to 0$. Thus it suffices to constrain our attention to $C \in [M_1, M_2]$ for some $M_1, M_2 > 0$ depending only on $D_\lambda$.

However, $J_M(g, z_{i,j}, D_\lambda) - \inf J_M(h, z_{i,j}, D_\lambda) > 0$ (note that $\inf J_M(h, z_{i,j}, D_\lambda) = \inf J_M(h, \mathcal{X}, D_\lambda)$) for all $C \in [M_1, M_2]$. Since this is a continuous function of $C$ over a compact set, it must obtain a minimum greater than 0, and we may choose $\epsilon_0$ to be this minimum (rescaled by a factor of $\Omega(1/N^2)$), thereby finishing the proof. $\square$

## C.2 Proofs of Propositions 3.4 and 3.5

**Proposition 3.4.** [Mixup Gradient Lower Bound] Let $y$ be any class such that $\Delta_y \geq \log k$, and suppose that both $\langle w_y, v_{y,1} \rangle \geq 0$ and the cross-class orthogonality condition $\langle w_s, v_{u,\ell} \rangle = 0$ hold for all $s \neq u$ and $\ell \in [2]$. Then we have with high probability that:

$$\langle -\boldsymbol{\nabla}_{w_y} J_{MM}(g, \mathcal{X}), v_{y,2} \rangle \geq \Theta\left(\frac{1}{k^2}\right) \tag{3.5}$$

*Proof.* The idea of proof will be to analyze the gradient correlation with $v_{y,1} - v_{y,2}$, and either show that this is significantly negative or, in the case where it is not, the gradient correlation with $v_{y,2}$ is still significant. Firstly, using the cross-class orthogonality assumption and Calculation A.10, we can compute:

$$\langle -\boldsymbol{\nabla}_{w_y} J_{MM}(g, \mathcal{X}), v_{y,1} - v_{y,2} \rangle = \frac{1}{N^2} \sum_{i \in N_y} \sum_{j \notin N_y} \left(1 - 2\phi^y((g(z_{i,j})))\right)\left(\beta_i - \frac{1}{2}\right)$$
$$+ \frac{1}{N^2} \sum_{i \in N_y} \sum_{j \notin N_y} \left(1 - \phi^y((g(z_{i,j})))\right)\left(\beta_i - \frac{1}{2}\right) \tag{C.5}$$

Where above we used $N_y$ to indicate the indices corresponding to class $y$ data points (as we do in the proofs of the main results). Now using concentration of measure for bounded random variables and the fact that $N$ is sufficiently large, we have from Equation C.5 that with high probability (and with $\text{poly}(k)$ representing a very large polynomial in $k$):

$$\langle -\boldsymbol{\nabla}_{w_y^{(t)}} J_{MM}(g, \mathcal{X}), v_{y,1} - v_{y,2} \rangle \leq \Theta\left(\frac{1}{k^2}\right) \sum_{s \neq y} \mathbb{E}_{\beta_s, \beta_y}[A_1(\beta_s, \beta_y)(\beta_y - 1/2)]$$
$$+ \Theta\left(\frac{1}{k^2}\right) \mathbb{E}_{\beta_y}[A_2(\beta_y)(\beta_y - 1/2)] + O(1/\text{poly}(k)) \tag{C.6}$$

Where we define the functions $A_1$ and $A_2$ as:

$$A_1(\beta_s, \beta_y) \triangleq 1 - 2\phi^y((g_t(z_{y,s}))) \tag{C.7}$$
$$A_2(\beta_s, \beta_y) \triangleq 1 - \phi^y((g_t(z_{y,s}))) \tag{C.8}$$

With $z_{y,s}$ being a random variable denoting the sum of a class $y$ point and a class $s$ point (distributed according to Definition 4.1). Note that Equations C.7 and C.8 are not abuses of notation - the functions $A_1$ and $A_2$ depend only on the random variables $\beta_s$ and $\beta_y$, since we can ignore the cross-class correlations due to orthogonality.

Let us immediately observe that the first two terms (the expectation terms) in Equation C.6 are bounded above by 0. This is due to the fact that $\beta_y - 1/2$ is a symmetric, centered random variable and the functions $A_1$ and $A_2$ are monotonically decreasing in $\beta_y$. We will focus on showing that the first term is significantly negative, as that will be sufficient for our purposes.

Now we let $\Xi_y = \langle w_y, v_{y,1} \rangle + \langle w_y, v_{y,2} \rangle$ and perform the transformation $g_t^s \leftarrow g_t^s - \Xi_y/2$ for all $s \in [k]$ (note this doesn't change the value of the softmax outputs). Under this transformation we have that $g_t^y(z_{y,s}) = (\beta_y - 1/2)\Delta_y$, which isolates the gap term $\Delta_y$.

For further convenience, let us use $\Lambda_s \triangleq \sum_{u \neq y} \exp(g_t^u(z_{y,s}))$, and observe that $\Lambda_s$ depends only on $\beta_s$ due to orthogonality. Using the change of variables $\tilde{\beta}_y = \beta_y - 1/2$ we can then compute the expectation in the first term of Equation C.6 as:

$$
\mathbb{E}_{\beta_s, \beta_y} \left[ A_1(\beta_s, \beta_y)(\beta_y - 1/2) \right] = \frac{1}{0.64} \int_{0.1}^{0.9} \int_{0.1}^{0.9} \frac{\Lambda_s - \exp(g_t^y(z_{y,s}))}{\Lambda_s + \exp(g_t^y(z_{y,s}))} (\beta_y - 1/2) \, d\beta_y \, d\beta_s
$$

$$
= \frac{1}{0.64} \int_{-0.4}^{0.4} \int_{-0.4}^{0.4} \frac{\Lambda_s - \exp\left(\tilde{\beta}_y \Delta_y\right)}{\Lambda_s + \exp\left(\tilde{\beta}_y \Delta_y\right)} \tilde{\beta}_y \, d\tilde{\beta}_y \, d\tilde{\beta}_s \tag{C.9}
$$

We will focus on the inner integral in Equation C.9. Using the symmetry of $\tilde{\beta}_y$, we have that:

$$
\mathbb{E}_{\beta_y} \left[ A_1(\beta_s, \beta_y)(\beta_y - 1/2) \right] = \frac{1}{0.8} \int_{-0.4}^{0.4} \frac{\Lambda_s - \exp\left(\tilde{\beta}_y \Delta_y\right)}{\Lambda_s + \exp\left(\tilde{\beta}_y \Delta_y\right)} \, d\tilde{\beta}_y
$$

$$
= -\frac{1}{0.8} \int_0^{0.4} \left( \frac{\Lambda_s - \exp\left(-\tilde{\beta}_y \Delta_y\right)}{\Lambda_s + \exp\left(-\tilde{\beta}_y \Delta_y\right)} - \frac{\Lambda_s - \exp\left(\tilde{\beta}_y \Delta_y\right)}{\Lambda_s + \exp\left(\tilde{\beta}_y \Delta_y\right)} \right) \tilde{\beta}_y \, d\tilde{\beta}_y
$$

$$
= -\frac{1}{0.8} \int_0^{0.4} \frac{\Lambda_s \left( \exp\left(\tilde{\beta}_y \Delta_y\right) - \exp\left(-\tilde{\beta}_y \Delta_y\right) \right)}{\Lambda_s^2 + \Lambda_s \left( \exp\left(\tilde{\beta}_y \Delta_y\right) + \exp\left(-\tilde{\beta}_y \Delta_y\right) \right) + 1} \tilde{\beta}_y \, d\tilde{\beta}_y
\tag{C.10}
$$

From our orthogonality assumption and the facts that $\Delta_y \geq \log k$ and $\langle w_y, v_{y,2} \rangle \geq 0$, we have that $\sum_{u \neq y,s} \exp(g_t^u(z_{y,s})) = O(1)$. Additionally, if we let:

$$
D_{y,s} = (C_{y,1} + C_{y,2}) - (C_{s,1} + C_{s,2})
$$

Then we get from Equation C.10 that:

$$
\mathbb{E}_{\beta_s, \beta_y} \left[ A_1(\beta_s, \beta_y)(\beta_y - 1/2) \right]
$$

$$
= -\int_{-0.4}^{0.4} \int_0^{0.4} \Theta \left( \frac{\tilde{\beta}_y \exp\left(\tilde{\beta}_s \Delta_s - D_{y,s}/2\right) \exp\left(\tilde{\beta}_y \Delta_y\right)}{\exp\left(2\tilde{\beta}_s \Delta_s - D_{y,s}\right) + \exp\left(\tilde{\beta}_s \Delta_s - D_{y,s}/2\right) \exp\left(\tilde{\beta}_y \Delta_y\right)} \right) d\tilde{\beta}_y \, d\tilde{\beta}_s
\tag{C.11}
$$

Now we consider two cases. First, suppose there exists a set $U \subset [-0.4, 0.4] \times [0, 0.4]$ with probability measure at least 0.01 such that for all $(\tilde{\beta}_s, \tilde{\beta}_y) \in U$:

$$
\tilde{\beta}_s \Delta_s - D_{y,s}/2 \leq \tilde{\beta}_y \Delta_y
$$

Then we immediately have that Equation C.11 is $\Theta(1)$. On the other hand, if this is not the case, one can see that $\mathbb{E}_{\beta_s, \beta_y}[A_1(\beta_s, \beta_y)(1 - \beta_y)] = \Theta(1)$, so we are done. □

**Proposition 3.5.** [ERM Gradient Upper Bound] For every $y \in [k]$, assuming the same conditions as in Proposition 3.4, if $\Delta_y \geq C \log k$ for any $C > 0$ then with high probability we have that:

$$
\langle -\nabla_{w_y} J(g, \mathcal{X}), v_{y,2} \rangle \leq O \left( \frac{1}{k^{0.1C-1}} \right)
\tag{3.6}
$$

*Proof.* From the facts that $\langle w_y, v_{y,2} \rangle \geq 0$ and $\Delta_y \geq C \log k$, we have that $g^y(x_i) \geq \beta_{y,i} C \log k$ for every $i \in N_y$ (where $\beta_{y,i}$ represents the coefficient in front of $v_{y,1}$ in $x_i$). Since $\beta_{y,i} \in [0.1, 0.9]$, we immediately have the result from the logic of Claim B.2 and Calculation A.9. □

## C.3 Proof of Proposition 4.4

**Proposition 4.4.** There exists a $\mathcal{D}$ satisfying all of the conditions of Definition 4.1 and Assumption 4.3 such that with probability at least $1 - k^2 \exp\left(\Theta(-N/k^2)\right)$, for any classifier $h : \mathbb{R}^{Pd} \to \mathbb{R}^k$ of the form $h^y(x) = \sum_{p \in [P]} \langle w_y, x^{(p)} \rangle$ and any $\mathcal{X}$ consisting of $N$ i.i.d. draws from $\mathcal{D}$, there exists a point $(x, y) \in \mathcal{X}$ and a class $s \neq y$ such that $h^s(x) \geq h^y(x)$.

*Proof.* For hyperparameters, we can choose $\delta_1 = \delta_2 - \delta_1 = 1$, $\delta_3 = \delta_4 = k^{-1.5}$ while being consistent with Assumption 4.3. For the distribution $\mathcal{D}$, for each point $(x_i, y_i)$, we sample a class $s \in [k] \setminus y_i$ uniformly and choose it to be the single class used in the feature noise patches for $x_i$. This clearly falls within the scope of Definition 4.1.

Now in $N$ i.i.d. samples from $\mathcal{D}$ as specified above, we have with probability at least $1 - k^2 \exp\left(\Theta(-N/k^2)\right)$ (Chernoff bound, as in Claim B.1) that a sample with each possible pair $y$, $s \in [k]$ of signal and noise classes exists. Suppose now that there exist classes $y$, $u \in [k]$ such that $u \notin \operatorname{argmax}_{s \in [k]} \langle w_s, v_{y,1} + v_{y,2} \rangle$. This necessarily implies that $u \neq \max_{s \in [k]} g^s(x)$ for all points $x$ with label $u$ but having feature noise class $y$ (since the order of the sum of the feature noise is $\Theta(\sqrt{k})$), which gives the result.

On the other hand, if there does not exist such a class pair $y$, $u$, then we are also done as that implies all of the weight-feature correlations are the same. $\qquad\square$

