# OpenReview forum: "Provably Learning Diverse Features in Multi-View Data with Midpoint Mixup"
_ICLR.cc/2023/Conference — Submitted to ICLR 2023_

### Official Review · Reviewer_9dJZ · 2022-10-19

**Confidence:** 3
**Clarity, Quality, Novelty And Reproducibility:** 1. Overall the paper writing is clear…
**Correctness:** 3
**Technical Novelty And Significance:** 3
**Empirical Novelty And Significance:** 3
**Recommendation:** 8

**Strength And Weaknesses:**

Strength:
1. On top of Allen-Zhu and Li (2021), the authors further decompose the input into multiple orthonormal feature views. This allows the authors to analyze the contribution of each individual feature and their gradient correlations.
2. Although the change of data settings (section 3.1) and assumptions (3.10) is simple, the authors provide significant efforts in proving the feature learning under the mixup loss.

Weakness:
* Theory:
1) In Eq. B.34, do you need a lower bound or dependence on the training iterations, such that the feature correlations will not be over-corrected?
2) Could you add more intuitive explanations of the meaning of function f in Assumption 3.10? Is it possible to analytically prove the monotonicity of f?

* Experiments:
1) Thm 3.9 and 3.11 did not directly imply better test accuracy (generalization). Instead, is it possible to empirically verify the one or all feature(s) (v) learned (defined by 3.8) by the Midpoint Mixup?
2) It would be better to add variances in Table 1.
3) Many test errors in Table 1 are far worse than random guess (90% error rate). From Eq. 4.1, I did not see any noise is introduced. What is the reason for this large test error?

**Summary Of The Paper:**

This work adopts the feature learning framework by Allen-Zhu and Li (2021), and further introduces latent orthogonal feature views. The authors prove that under certain assumptions on features, a two-layer smooth-ReLU network can learn all features with midpoint mixup, whereas ERM can only learn one feature.

**Summary Of The Review:**

In general, I believe this work contributes significantly to the feature learning topic.
I hope the authors could fix my concerns above and add more clarifications.

---

> ### Author Response · Authors · 2022-11-12
> **Response to Reviewer 9dJZ**
>
> We would like to thank Reviewer 9dJZ for their kind words and helpful feedback, and appreciate that the reviewer found the work to be a significant contribution. We attempt to address all of the expressed concerns below.
>
> 1. Theory
>     * The bound in Equation B.34 is actually tight (asymptotically), thank you for mentioning and carefully reading that - we have fixed it to be equality in the revision (you can see in the original proof in the discussion under Equation B.41 that explains why).
>     * We totally acknowledge the unintuitive nature of Assumption 3.10 in the original version of the paper. In the revision, we found an approach that avoided having to prove Assumption 3.10, so our theory no longer depends on it (which is a significant strengthening).
>
> 2. Experiments
>     * Indeed, our theorems did not directly imply better generalization (unless there is some known distributional shift). We also completely agree with all reviewers that our original experimental setup was too noisy; we have changed it entirely as per the general comment we made above. In the context of image datasets, it is a bit tricky to verify the feature learning aspect in the sense that one needs a notion of features with respect to the images. However, if we assume such latent features exist, we can attempt to verify this by considering training data that is modified to be concatenations of images from different classes. In this way, it is possible to achieve perfect perfect training accuracy via learning spurious features (features from other classes) for each class. We hope with this in mind, the interpretation of experimental results makes more sense. Another limitation is that our theory is in the asymptotic regime, and it is computationally infeasible to train the true Midpoint Mixup loss (requires seeing all $N^2$ Mixup points per epoch).
>     * We have added standard deviations (when significant) to the test errors in Table 1. They are also shown as shaded areas in the plots in Figure 1.
>     * The test errors were originally very poor due to the fact that we *removed* the added orthonormal features from training at test time. We realized upon further analysis that using these orthonormal spurious features in training made minimizing the training loss too easy (i.e. overfitting to the spurious features), which led to sharp degradations in the test accuracy. This is no longer an issue in the new experimental setup.

---

> > ### Comment · Reviewer_9dJZ · 2022-12-05
> > **Thanks for the authors' response**
> >
> > I read through the authors' response and I have no further concerns. I keep my score.

---

### Official Review · Reviewer_uhyy · 2022-10-23

**Confidence:** 4
**Correctness:** 2
**Technical Novelty And Significance:** 2
**Empirical Novelty And Significance:** 2
**Recommendation:** 3

**Clarity, Quality, Novelty And Reproducibility:**

The problem itself seems to be novel, but the theoretical results are unclear. Below I wrote some questions to clarify them.

* I couldn't fully get the proof of Lemma 2.1. Why does the “existence of well-defined h simultaneously optimizing each term of eq.2.3” implies “RHS of (2.4) is achieving the minimum”, so that we have “$g^{y_i} (z_{i,j}) = g^{y_j} (z_{i,j})$ for all $z_{i,j}$”?

* I couldn't get the sentence "J(g, X) by just taking $⟨w_y, v_{y,1}⟩ \rightarrow \infty$ for every class y" in the paper. I thought setting $w_c = v_{y,1}$ was enough to make the empirical loss = 0?

* I couldn't get what footnote 1 means.

* Sec.3.1 title is “linear model” and Sec.3.2 title is “multi-view data setup”. But it seems like Def.3.1 in Sec.3.1 is also multi-view. What is the criteria differentiating Sec.3.1 and 3.2? Maybe because Sec.3.2 for general model, not limited to linear model? If so, the section name is misleading.


I am also adding minor comments
* The first sentence of Definition 3.4 is “Identically to Definition 3.4”. Maybe typo for “Identically to Definition 3.1”?
* Typo: themself → themselves

**Strength And Weaknesses:**

- Strength
    - The direction of analyzing mid-point mixup on mulit-view data is new and interesting.

- Weakness

    - Theory
        - Why is the assumption of Lemma 2.1 (z_{i,j} is unique for all i,j pair) true in practice? Simple 2-dimensional dataset where each class follows Gaussian distribution cannot follow this assumption. (how about high-dimensional case?)
        - It seems like Lemma 2.1 is important to show that Midpoint Mixup is not poor, but the proof of Lemma 2.1 is not fully understood on my side. (I put the details in the questions section)

    - Experiments
        - This paper shows that when L is large, Mid-point Mixup is having less error than ERM. Of course if L is large, it will go to the ideal multi-view dataset this paper is assuming, but then why don’t we even start with real data? Instead, making a synthetic multi-view data to show the effectiveness of the theory seems more reasonable. I personally feel the experiments in this paper is similar to just “simulation” instead of real-world experiments.
        - In Table 1, the authors are comparing “test error 96%” versus “test error 97%”, when we have 100 classes. What does it mean to increase 1% test error from “nearly random guessing”? Does these empirical results have any positive impact on the practical deployment of mid-point mixup?

    - Writing
        - The assumptions and results are less organized (thus I have many questions below). The definition 3.4 has no intuitive explanation or visualization.

**Summary Of The Paper:**

This paper focuses on mid-point mixup (a variant of mixup) and provides theoretical analysis on the effect of using mid-point mixup on multi-view data. The theoretical result is specific to the setting of the 2-layer convolutional networks and the datasets where samples at each class have 2 features. Under this setting, this paper shows that empricial risk minimization learn only single feature at each class, while mid-point mixup learns both features at each class.

**Summary Of The Review:**

This paper focuses on an interesting problem, but has much room for improvement in terms of organizing the results, validating the assumption, and explaining how the experimental results are helpful in practical settings.

---

> ### Author Response · Authors · 2022-11-12
> **Response to Reviewer uhyy**
>
> We would like to thank Reviewer uhyy for their helpful feedback and try to address the expressed concerns below.
>
> 1. Theory
>     * We have reorganized the paper and thus reformulated Lemma 2.1 into Lemma 3.2, which is in a different context and hopefully much easier to interpret. Our goal with Lemma 2.1 was to show why Midpoint Mixup is particularly interesting, and we believe the new Lemma 3.2 better demonstrates that. With that being said, we answer the original questions as stated below.
>     * We would like to point out a minor correction in the interpretation of the condition in the original Lemma 2.1: we simply require that two midpoints cannot coincide if their classes also do not coincide (i.e. a class 1-2 midpoint cannot coincide with a class 3-4 midpoint). We are not fully sure what is meant by the Gaussian data example - if this is to mean data points that are sampled from a Gaussian (or high-dimensional Gaussian), then with probability 1 _any pair_ of midpoints will not coincide. In fact this is true for points sampled from _any_ continuous distribution, regardless of dimensionality. The point of this condition is just to guarantee that there is no cross-class collinearity, as it has been pointed out previously that when such collinearity exists one cannot hope to succeed using Mixup (even for more canonical mixing distributions - see the discussion in [1]).
>     * To clarify your question regarding the proof of the original Lemma 2.1: basically the line you have mentioned points out that we can construct a function $h$ such that $h^{y_i}(z_{i, j}) = h^{y_j}(z_{i, j})$ because we know that there is no other midpoint that coincides with $z_{i, j}$. We are saying nothing about the behavior of this function elsewhere on its domain, we are just pinning it down on the Midpoint Mixup points, which we can do because none of them conflict with one another - hopefully that clarifies things.
>
> 2. Experiments
>     * In our original experiments, we considered increasing levels of $L$ to showcase the phenomenon with different amounts of added features; while we originally thought these experiments to be relatively interesting in the 1 and 2 added feature regime, we realize now that the experimental setup was rather confusing and unintuitive, and have thus redone the experiments entirely (as detailed in the general comment). The new experiments focus on concatenated examples from the original datasets, where a model could achieve good training performance for a class by learning features associated with a different class. The test data remains unchanged, so as to verify whether the features for each class were actually learned. This leads to much less noisy error curves and more reasonable final errors.
>     * With regards to synthetic experiments, we do provide an implementation of a multi-view data setup mirroring our theory as close as possible, but we find that with the hyperparameter settings in our theory it is not feasible to train in practice (the gradients are vanishingly small, see the README in the attached code).
>
> 3. Writing
>     * We completely acknowledge that Assumption 3.10 could be much better motivated, and we apologize that this was not done in the original version of the paper. In our revision, we have improved the theory to no longer depend on Assumption 3.10 (see the general note for a brief mention of the ideas), and we believe this significantly strengthens the value of our theory.
>     * With regards to Definition 3.4 (now Definition 4.1 in the revision) we agree that it is a very technical definition, but as it is intended to be a generalization of Definition 3.1 we intended Section 3.1 to be motivation for Definition 3.4. We are happy to clarify any specific questions you may have about the definition; the intuition one could use is to consider "images" composed of patches (over which one applies a convolution), and these patches consist of different "features" (orthogonal vectors) with different weights applied to them.
>
> 4. Other Clarifications
>     * In the case of the linear model, it is not sufficient just to have $w_y = v_{y, 1}$ to minimize the empirical cross-entropy, as one needs to take the scaling to $\infty$ so that the softmax output approaches 1 for the correct class.
>     * The original footnote 1 was intended to mean that we do not need a dependency structure as strong as mirrored coefficients to benefit from Midpoint Mixup; mainly, the idea is that for Midpoint Mixup the optimal strategy is to predict as close to 1/2 for each class as possible on the Midpoint Mixup points, and to do so we want to minimize the variance of the output.
>     * We have clarified the section titles to make it clearer that the linear example handles a limited version of multi-view data while our main results handle a more general version of multi-view data.
>     * Thank you for catching the typos, we have corrected them in the revision.
>
> [1] https://arxiv.org/abs/2110.07647

---

### Official Review · Reviewer_Pxqu · 2022-10-25

**Confidence:** 4
**Correctness:** 4
**Technical Novelty And Significance:** 3
**Empirical Novelty And Significance:** 3
**Recommendation:** 6

**Clarity, Quality, Novelty And Reproducibility:**

The clarity is decent. the novelty is partially limited because the proof/data framework is largely inherited from prior work, while the original part (such as assumption 3.10) lacks enough explanation.

**Strength And Weaknesses:**

Strength: Most of the paper is clearly written and it provides a rigorous analysis of the subject it studies. Mixup is an interesting data augmentation technique and tries to explain why it can help in improving the feature learning of neural nets. The theory result goes beyond the previous works that are based on linear models (as they are based on nonlinear neural networks) and is reasonable due to the feature learning technique it uses. It might be an important paper if the authors can fix the issues I find below.

Weaknesses:
1. The authors did not provide sufficient motivation why they specifically study Midpoint Mixup rather than the general mixup augmentations. Moreover, they did not provide an empirical comparison between the original mixup and the midpoint mixup, making it hard to justify their choice of study. A potential improvement is to explain the technical difficulty of analyzing the original mixup, and justifies that the midpoint mixup is competitive (or particularly interesting).
2. Assumption 3.10, which seems to be the key assumption to migrate the proof technique in [2] to the setting of this paper, is very technical, without intuitive explanation, and without support from empirical evidence. It is hard to make sense of this assumption as assembling any real-world structure. This weakens the validity of the proposed explanation as the theory result could be just an artificial construction.
3. The experimental results in this paper are not informative, as many of their metrics are too low. I do not understand why this paper constructs such special data by mixing images using Dirichlet distribution (see equation 4.1 and Fig 1) to compare ERM with Midpoint-Mixup. Moreover, the test errors often exceed 60%-90% for experiments on CIFAR-10/100 when L>1 (that is over the special dataset rather than the original Cifar10/100), meaning that the model struggles a lot to predict the mixup label. I don't think these experiments are sufficient to support the authors' choice of study or their theoretical claims.


[1] Ruoqi Shen, Sebastien Bubeck, Suriya Gunasekar, Data Augmentation as Feature Manipulation
[2] Zeyuan Allen-Zhu, Yuanzhi Li, Towards Understanding Ensemble, Knowledge Distillation and Self-Distillation in Deep Learning

**Summary Of The Paper:**

This paper studies a specific instance of Mixup data augmentation: Midpoint Mixup. It provides a theoretical characterization of the learning dynamics of neural networks with mixup-augmented data, and proved how it can improve feature learning by helping the neural network pick up diverse features, under the multi-view data framework proposed by Allen-Zhu and Li.


[1] Zeyuan Allen-Zhu, Yuanzhi Li, Towards Understanding Ensemble, Knowledge Distillation and Self-Distillation in Deep Learning

**Summary Of The Review:**

This paper studies an interesting instance of Mixup augmentation and obtained some decent theory results with the help of an established theoretical framework, but the authors are not good at justifying their choice of study and their theoretical claims, and they fail at explaining the key technical ingredient of the paper, as well as how well their assumptions can connect to practical scenarios.

---

> ### Author Response · Authors · 2022-11-12
> **Response to Reviewer Pxqu**
>
> We would like to thank Reviewer Pxqu for taking the time to review our paper and providing helpful comments, and we appreciate the kind remarks regarding the potential importance of the work. We hope to address all of the expressed concerns below.
>
> 1. Insufficient motivation for Midpoint Mixup.
>     * Upon reading your comment and reviewing our paper, we realized that the alignment lemma in our original paper (Lemma 2.1) did not adequately capture our reasons for studying Midpoint Mixup. We have now added significantly more exposition motivating the study of Midpoint Mixup in the section analyzing linear models on linearly separable data. Namely, Midpoint Mixup has two key benefits from a theoretical perspective: its global minimizers have the feature learning property we desire, and the structure of its gradients are not changing with each optimization iteration (due to a fixed mixing proportion). We now emphasize this, and also stress that we did not intend for Midpoint Mixup to be considered a practical alternative to regular Mixup (due to its high level of regularization), but rather intended it to be a helpful theoretical approximation that accentuates some useful behaviors of Mixup.
>
> 2. Assumption 3.10 is too technical and not sufficiently motivated.
>     * In carefully reviewing our proof details, we found a way around Assumption 3.10 - it has been removed, and our main theoretical result no longer relies on it (a significant improvement in their generality). We explain the reason we needed Assumption 3.10 and how we got around it in our general comment; we did not expect to be able to address this dependence on Assumption 3.10 during the reviewing period, but time away from the paper was helpful for us in approaching with new ideas, and we thank you for pointing it out as a significant weakness.
>
> 3. Insufficient/unrealistic experimental results.
>     * We definitely agree that, as originally written, the experiments were confusing and some of them had errors that were too high to be reasonably comparable. We do want to point out one correction to the original review comment, which is that the performance reported was over the *unmodified* test data for each dataset, which is why test errors were so high.
>
>         We have completely rewritten the experiments section and redone our experimental setup; we found upon analyzing our experiments further that additive noise in the form of orthonormal feature vectors that are fixed per class makes the training problem too easy (training error goes to 0 quickly) while test performance suffered significantly in the absence of these features. We have thus focused on a new setup using concatenation of images from different classes, we find this to be a more intuitive way of mixing features present in the original data. We also realized that the framing of our original experiments did not clearly reflect our goal; in our new experiments, we compare Mixup (with standard hyperparameters), Midpoint Mixup, and ERM. Our experiments show that regular Mixup outperforms ERM in all cases when these spurious features are removed, while Midpoint Mixup empirically also captures a lot of this benefit (and improves as the number of classes increases). Once again, our goal was not to propose Midpoint Mixup as a practical alternative, but rather as a theoretical alternative with reasonably close practical performance.

---

> > ### Comment · Reviewer_Pxqu · 2022-12-08
> > **Response to authors.**
> >
> > Thanks for the response, I am glad to see these improvements that are made in such a short time. I am satisfied with most of the changes that addressed my concerns in the initial review, and I intend to raise my score to borderline acceptance to reflect these improvements.
> >
> > However, the reason that I did not raise my score to acceptance is that I am still not fully motivated by the story in the motivation section. Midpoint-Mixup data augmentation is an empirically interesting discovery with many mysteries around it, but it seems the authors are more interested in the theoretical motivations (which can be limited by data and modeling choices). And moreover, this paper focuses too much on the "results", namely convergence guarantees. It is more interesting to study the empirical properties of Midpoint-Mixup and use theory to unveil its mysteries, rather than a purely theoretical statement that could be artificial. Thus I vote for borderline acceptance.

---

> > > ### Author Response · Authors · 2022-12-08
> > > **Response to Reviewer Pxqu**
> > >
> > > Thank you for taking the time to read our response and updating your score accordingly. We are happy to see that the revisions have addressed most of your concerns. We also completely agree that Midpoint Mixup has many mysterious and interesting empirical properties that are beyond the scope of our paper; for example, it is not obvious why Midpoint Mixup is able to train so quickly in the mini-batch setting given that, in expectation, it only sees a single true data point per epoch (there are $N^2$ possible mixing pairs and a true data point occurs only when one mixes a point with itself). As we state in the paper, we believe that a further exploration of these phenomena is a fruitful direction for future work.
> > >
> > > To shed some light on why we focus on theoretical motivations for Midpoint Mixup, we stress that the general motivation for our paper (as outlined in the introduction) was to address the following question: "How can we theoretically explain the benefits of optimizing the Mixup objective in the context of training neural networks?" Towards this end, we introduce and motivate Midpoint Mixup as a useful _lens_ into the behavior of Mixup, and we prove multiple properties in Section 3 of our paper showing why Midpoint Mixup is a nicer objective to study than standard Mixup.
> > >
> > > Furthermore, we feel that good theoretical motivation is basically necessary when introducing Midpoint Mixup because, a priori, a practitioner would expect it to be an extreme setting of Mixup that introduces too much regularization (referring back to the simple calculation above, mini-batch training using Midpoint Mixup consists almost entirely of midpoints of different data points).
> > >
> > > With regards to the possible artificiality of the theory, we definitely acknowledge that the theoretical settings we consider are still far from the experimental setting of training deep networks with complex architectures on realistic datasets. Indeed, this is an issue that almost anyone working in this area has to acknowledge. However, our hope is that the key insights that we develop in these theoretical approximations are predictive of the more complicated empirical scenarios.
> > >
> > > In our case, the takeaway of our paper is that Midpoint Mixup (and, empirically, regular Mixup) can help learn multiple features present in the data, and we establish the reason behind this as a "correction" effect that occurs in the optimization dynamics (Proposition 3.4, Theorem 4.7). We show this correction effect in the linear model setting, and then in the 2-layer CNN setting, and the hope is that given we were able to extend from linear models to simple non-linear models, with stronger mathematical tools it will be possible to extend the same idea to even more complicated non-linear settings.
> > >
> > > Finally, we believe that the ideas used in the proofs of our main results have value in and of themselves, even if one believes the theory is done in a stylized setting. For those working in theory, the gradient correction effect that we show may have applications in the analysis of other data augmentation techniques. For practitioners, one could use our ideas to reason about whether Mixup is a potentially useful technique to try in their setting - if one expects a dataset to have multiple predictive features per class, then our theory and experiments suggest Mixup should have value. Given the ever-increasing costs of training state-of-the-art models, we believe that this kind of reasoning (which avoids having to run an experiment to check) will become more and more valuable.
> > >
> > > We hope the above provides more clarity on why we made the decisions that we did in our paper, and we are happy to answer any further questions you may have.

---

### Author Response · Authors · 2022-11-12
**Overview of Significant Revisions**

We apologize for responding with some delay and thank all of the reviewers for their feedback; we found it very helpful, as it made clear that we did not correctly convey some of the goals of our work in the main body of the paper. We have made a few significant revisions to the paper, which we outline below.

1. Significantly more motivation for Midpoint Mixup, and comparison with standard Mixup settings.
    * Upon reading the reviews we realized that the original "alignment lemma" (Lemma 2.1 in the first version) in our paper did not adequately motivate Midpoint Mixup, as we introduced it prior to discussing the notion of feature-learning. We have remedied this by adding significantly more exposition regarding why we consider Midpoint Mixup, and transforming the alignment lemma into the feature learning context via Lemma 3.2 in the new write-up. We have also included a contrasting result (Proposition 3.3) for regular Mixup to show that it does not have an equivalent property.
    * The goal of our analysis of Midpoint Mixup was definitely what Reviewer Pxqu alluded to - namely, it is a setting of Mixup that allows for clean theoretical analysis due to some nice properties. We did not intend to insinuate that Midpoint Mixup is a better practical alternative than Mixup with regular settings, and we have now stressed this point throughout the paper (and in the experiments).

2. Improving theory by removing Assumption 3.10 from the original version of the paper.
    * This is perhaps the biggest improvement in our revision. We completely agree with the reviewers that Assumption 3.10 was a highly technical (and not intuitively explained) assumption that hampered the generality of our theory. We needed this assumption because of a difficulty in relating the linear analysis to the non-linear analysis that results from our activation function; we spent a long time trying to prove properties about this relationship (Assumption 3.10 originally) with little success. However, in carefully going over our proof, we found a different avenue of attack that did not require the aforementioned type of analysis, which allowed us to prove our main results for Midpoint Mixup without any additional assumptions. The key idea was to improve our strategy in the linear case to a strategy that handles each of the signal patches individually instead of in aggregate. One can see the revision (it is relatively short) of Part II of the Midpoint Mixup analysis in the appendix for details.

3. Rewriting experiments entirely to be much more comparable.
    * Upon reviewing the experimental results as well as the reviewers' feedback, we definitely agree we should have thought more carefully about our original experimental setup. While we still believe the original experiments showed some interesting robustness properties of Midpoint Mixup, we agree that many of the experiments were possibly inconclusive due to the high test errors.

        We have fixed this by rewriting the experiments section entirely, with new experiments focusing on what we hope is a much more intuitive way of introducing spurious features into the original datasets (concatenation with other images). We have also included a discussion of alternate experimental setups and why we chose the new setup, and have also removed many extraneous experiments that were not particularly informative in the original paper. Additionally, we have further highlighted that the goal of our experiments was to show that Midpoint Mixup reasonably captures the benefits of regular Mixup by including performance results for a standard setting of Mixup. Our goal was never to claim that Midpoint Mixup is a superior practical alternative, the take-away message we desired was that Midpoint Mixup is a useful theoretical setting for analysis, and that it can perform comparably to Mixup in experimental settings.

---

### Decision · Program_Chairs · 2023-01-20

**Decision:**

Reject

**Justification For Why Not Higher Score:**

Reviewers have unaddressed concerns on the paper, in particular about the experimental setup being impractical (test errors in the range of 90%) and less connected with theory. The authors have also made significant revisions to many parts of the original manuscript and there is a concern that this will require another careful review of the paper.

**Justification For Why Not Lower Score:**

N/A

**Metareview: Summary, Strengths And Weaknesses:**

The paper analyzes mid-point mixup, specifically the learning dynamics of neural networks with midpoint mixup-augmented data, and show how it can improve feature learning by helping the neural network learn diverse features, under the multi-view data framework proposed by Allen-Zhu and Li (2021). Authors also experiment with synthetic features augmented datasets that show that midpoint mixup outperforms ERM. The paper is the first to analyze mixup for nonlinear models and from a multiview feature learning perspective. However, reviewers have unaddressed concerns on the paper, in particular about the experimental setup being impractical (test errors in the range of 90%) and less connected with theory. The authors have also made significant revisions to many parts of the original manuscript and there is a concern that this will require another careful review of the paper. Authors should address the concerns about the alignment between theory and experiments and submit to a future venue.